# Deep Hyperalignment

**Muhammad Yousefnezhad, Daoqiang Zhang**
College of Computer Science and Technology
Nanjing University of Aeronautics and Astronautics
`{myousefnezhad,dqzhang}@nuaa.edu.cn`

## Abstract

This paper proposes Deep Hyperalignment (DHA) as a regularized, deep extension, scalable Hyperalignment (HA) method, which is well-suited for applying functional alignment to fMRI datasets with nonlinearity, high-dimensionality (broad ROI), and a large number of subjects. Unlink previous methods, DHA is not limited by a restricted fixed kernel function. Further, it uses a parametric approach, rank-$m$ Singular Value Decomposition (SVD), and stochastic gradient descent for optimization. Therefore, DHA has a suitable time complexity for large datasets, and DHA does not require the training data when it computes the functional alignment for a new subject. Experimental studies on multi-subject fMRI analysis confirm that the DHA method achieves superior performance to other state-of-the-art HA algorithms.

## 1 Introduction

The multi-subject fMRI analysis is a challenging problem in the human brain decoding [1–7]. On the one hand, the multi-subject analysis can verify the developed models across subjects. On the other hand, this analysis requires authentic functional and anatomical alignments among neuronal activities of different subjects, which these alignments can significantly improve the performance of the developed models [1, 4]. In fact, multi-subject fMRI images must be aligned across subjects in order to take between-subject variability into account. There are technically two main alignment methods, including anatomical alignment and functional alignment, which can work in unison. Indeed, anatomical alignment is only utilized in the majority of the fMRI studies as a preprocessing step. It is applied by aligning fMRI images based on anatomical features of standard structural MRI images, e.g. Talairach [2, 7]. However, anatomical alignment can limitedly improve the accuracy because the size, shape and anatomical location of functional loci differ across subjects [1, 2, 7]. By contrast, functional alignment explores to precisely align the fMRI images across subjects. Indeed, it has a broad range of applications in neuroscience, such as localization of the Brain's tumor [8].

As the widely used functional alignment method [1–7], Hyperalignment (HA) [1] is an 'anatomy free' functional alignment method, which can be mathematically formulated as a multiple-set Canonical Correlation Analysis (CCA) problem [2, 3, 5]. Original HA does not work in a very high dimensional space. In order to extend HA into the real-world problems, Xu et al. developed the Regularized Hyperalignment (RHA) by utilizing an EM algorithm to iteratively seek the regularized optimum parameters [2]. Further, Chen et al. developed Singular Value Decomposition Hyperalignment (SVDHA), which firstly provides dimensionality reduction by SVD, and then HA aligns the functional responses in the reduced space [4]. In another study, Chen et al. introduced Shared Response Model (SRM), which is technically equivalent to Probabilistic CCA [5]. In addition, Guntupalli et al. developed SearchLight (SL) model, which is actually an ensemble of quasi-CCA models fits on patches of the brain images [9]. Lorbert et al. illustrated the limitation of HA methods on the linear representation of fMRI responses. They also proposed Kernel Hyperalignment (KHA) as a nonlinear alternative in an embedding space for solving the HA limitation [3]. Although KHA

can solve the nonlinearity and high-dimensionality problems, its performance is limited by the fixed employed kernel function. As another nonlinear HA method, Chen et al. recently developed Convolutional Autoencoder (CAE) for *whole brain* functional alignment. Indeed, this method reformulates the SRM as a multi-view autoencoder [5] and then uses the standard SL analysis [9] in order to improve the stability and robustness of the generated classification (cognitive) model [6]. Since CAE simultaneously employs SRM and SL, its time complexity is so high. In a nutshell, there are three main challenges in previous HA methods for calculating accurate functional alignments, i.e. nonlinearity [3, 6], high-dimensionality [2, 4, 5], and using a large number of subjects [6].

As the main contribution of this paper, we propose a novel kernel approach, which is called Deep Hyperalignment (DHA), in order to solve mentioned challenges in HA problems. Indeed, DHA employs deep network, i.e. multiple stacked layers of nonlinear transformation, as the kernel function, which is parametric and uses rank-$m$ SVD [10] and Stochastic Gradient Descent (SGD) [13] for optimization. Consequently, DHA generates low-runtime on large datasets, and the training data is not referenced when DHA computes the functional alignment for a new subject. Further, DHA is not limited by a restricted fixed representational space because the kernel in DHA is a multi-layer neural network, which can separately implement *any nonlinear function* [11–13] for each subject to transfer the brain activities to a common space.

The proposed method is related to RHA [2] and MVLSA [10]. Indeed, the main difference between DHA and the mentioned methods lies in the deep kernel function. Further, KHA [3] is equivalent to DHA, where the proposed deep network is employed as the kernel function. In addition, DHA can be looked as a multi-set regularized DCCA [11] with stochastic optimization [13]. Finally, DHA is related to DGCCA [12], when DGCCA is reformulated for functional alignment by using regularization, and rank-$m$ SVD [10].

The rest of this paper is organized as follows: In Section 2, this study briefly introduces HA method. Then, DHA is proposed in Section 3. Experimental results are reported in Section 4; and finally, this paper presents conclusion and pointed out some future works in Section 5.

## 2 Hyperalignment

As a training set, preprocessed fMRI time series for $S$ subjects can be denoted by $\mathbf{X}^{(\ell)} = \left\{ x_{mn}^{(\ell)} \right\} \in \mathbb{R}^{T \times V}, \ell = 1{:}S, m = 1{:}T, n = 1{:}V$, where $V$ denotes the number of voxels, $T$ is the number of time points in units of TRs (Time of Repetition), and $x_{mn}^{(\ell)} \in \mathbb{R}$ denotes the functional activity for the $\ell$-*th* subject in the $m$-*th* time point and the $n$-*th* voxel. For assuring temporal alignment, the stimuli in the training set are considered time synchronized, i.e. the $m$-*th* time point for all subjects illustrates the same simulation [2, 3]. Original HA can be defined based on Inter-Subject Correlation (ISC), which is a classical metric in order to apply functional alignment: [1-4, 7]

$$\max_{\mathbf{R}^{(i)}, \mathbf{R}^{(j)}} \sum_{i=1}^{S} \sum_{j=i+1}^{S} \mathrm{ISC}(\mathbf{X}^{(i)}\mathbf{R}^{(i)}, \mathbf{X}^{(j)}\mathbf{R}^{(j)}) \equiv \max_{\mathbf{R}^{(i)}, \mathbf{R}^{(j)}} \sum_{i=1}^{S} \sum_{j=i+1}^{S} \mathrm{tr}\left( \left(\mathbf{X}^{(i)}\mathbf{R}^{(i)}\right)^{\top} \mathbf{X}^{(j)}\mathbf{R}^{(j)} \right) \quad (1)$$

$$\text{s.t.} \quad \left(\mathbf{X}^{(\ell)}\mathbf{R}^{(\ell)}\right)^{\top} \mathbf{X}^{(\ell)}\mathbf{R}^{(\ell)} = \mathbf{I}, \ell = 1{:}S,$$

where $tr()$ denotes the trace function, $\mathbf{I}$ is the identity matrix, $\mathbf{R}^{(\ell)} \in \mathbb{R}^{V \times V}$ denotes the solution for $\ell$-*th* subject. For avoiding overfitting, the constrains must be imposed in $\mathbf{R}^{(\ell)}$ [2, 7]. If $\mathbf{X}^{(\ell)} \sim \mathcal{N}(0, 1), \ell = 1{:}S$ are column-wise standardized, the ISC lies in $[-1, +1]$. Here, the large values illustrate better alignment [2, 3]. In order to seek an optimum solution, solving (1) may not be the best approach because there is no scale to evaluate the distance between current result and the optimum (fully maximized) solution [2, 4, 7]. Instead, we can reformulate (1) as a minimization problem by using a multiple-set CCA: [1–4]

$$\min_{\mathbf{R}^{(i)}, \mathbf{R}^{(j)}} \sum_{i=1}^{S} \sum_{j=i+1}^{S} \left\| \mathbf{X}^{(i)}\mathbf{R}^{(i)} - \mathbf{X}^{(j)}\mathbf{R}^{(j)} \right\|_{F}^{2}, \quad \text{s.t.} \left(\mathbf{X}^{(\ell)}\mathbf{R}^{(\ell)}\right)^{\top} \mathbf{X}^{(\ell)}\mathbf{R}^{(\ell)} = \mathbf{I}, \quad \ell = 1{:}S, \quad (2)$$

where (2) approaches zero for an optimum result. Indeed, the main assumption in the original HA is that the $\mathbf{R}^{(\ell)}, \ell = 1{:}S$ are noisy 'rotations' of a common template [1, 9]. This paper provides a detailed description of HA methods in the supplementary materials (`https://sourceforge.net/projects/myousefnezhad/files/DHA/`).

## 3 Deep Hyperalignment

Objective function of DHA is defined as follows:

$$\min_{\substack{\theta^{(i)}, \mathbf{R}^{(i)} \\ \theta^{(j)}, \mathbf{R}^{(j)}}} \sum_{i=1}^{S} \sum_{j=i+1}^{S} \left\| f_i\big(\mathbf{X}^{(i)}; \theta^{(i)}\big) \mathbf{R}^{(i)} - f_j\big(\mathbf{X}^{(j)}; \theta^{(j)}\big) \mathbf{R}^{(j)} \right\|_F^2 \tag{3}$$

$$\text{s.t.} \quad \left(\mathbf{R}^{(\ell)}\right)^\top \left( \left( f_\ell\big(\mathbf{X}^{(\ell)}; \theta^{(\ell)}\big) \right)^\top f_\ell\big(\mathbf{X}^{(\ell)}; \theta^{(\ell)}\big) + \epsilon \mathbf{I} \right) \mathbf{R}^{(\ell)} = \mathbf{I}, \quad \ell = 1{:}S,$$

where $\theta^{(\ell)} = \big\{ \mathbf{W}_m^{(\ell)}, \mathbf{b}_m^{(\ell)}, m{=}2{:}C \big\}$ denotes *all parameters in $\ell$-th deep network belonged to $\ell$-th subject*, $\mathbf{R}^{(\ell)} \in \mathbb{R}^{V_{new} \times V_{new}}$ is the DHA solution for $\ell$-th subject, $V_{new} \leq V$ denotes the number of features after transformation, the regularized parameter $\epsilon$ is a small constant, e.g. $10^{-8}$, and deep multi-layer kernel function $f_\ell\big(\mathbf{X}^{(\ell)}; \theta^{(\ell)}\big) \in \mathbb{R}^{T \times V_{new}}$ is denoted as follows:

$$f_\ell\big(\mathbf{X}^{(\ell)}; \theta^{(\ell)}\big) = \text{mat}\Big( \mathbf{h}_C^{(\ell)}, T, V_{new} \Big), \tag{4}$$

where $T$ denotes the number of time points, $C \geq 3$ is number of deep network layers, $\text{mat}(\mathbf{x}, m, n){:}\mathbb{R}^{mn} \to \mathbb{R}^{m \times n}$ denotes the reshape (matricization) function, and $\mathbf{h}_C^{(\ell)} \in \mathbb{R}^{T V_{new}}$ is the output layer of the following multi-layer deep network:

$$\mathbf{h}_m^{(\ell)} = \text{g}\Big( \mathbf{W}_m^{(\ell)} \mathbf{h}_{m-1}^{(\ell)} + \mathbf{b}_m^{(\ell)} \Big), \quad \text{where} \quad \mathbf{h}_1^{(\ell)} = \text{vec}\big(\mathbf{X}^{(\ell)}\big) \quad \text{and} \quad m = 2{:}C. \tag{5}$$

Here, $g{:}\mathbb{R} \to \mathbb{R}$ is a nonlinear function applied componentwise, $vec{:}\mathbb{R}^{m \times n} \to \mathbb{R}^{mn}$ denotes the vectorization function, consequently $\mathbf{h}_1^{(\ell)} = \text{vec}\big(\mathbf{X}^{(\ell)}\big) \in \mathbb{R}^{TV}$. Notably, this paper considers both $vec()$ and $mat()$ functions are linear transformations, where $\mathbf{X} \in \mathbb{R}^{m \times n} = \text{mat}\big(\text{vec}(\mathbf{X}), m, n\big)$ for any matrix $\mathbf{X}$. By considering $U^{(m)}$ units in the $m$-$th$ intermediate layer, parameters of distinctive layers of $f_\ell\big(\mathbf{X}^{(\ell)}; \theta^{(\ell)}\big)$ are defined by following properties: $\mathbf{W}_C^{(\ell)} \in \mathbb{R}^{T V_{new} \times U^{(C\text{-}1)}}$ and $\mathbf{b}_C^{(\ell)} \in \mathbb{R}^{T V_{new}}$ for the output layer, $\mathbf{W}_2^{(\ell)} \in \mathbb{R}^{U^{(2)} \times TV}$ and $\mathbf{b}_2^{(\ell)} \in \mathbb{R}^{U^{(2)}}$ for the first intermediate layer, and $\mathbf{W}_m^{(\ell)} \in \mathbb{R}^{U^{(m)} \times U^{(m\text{-}1)}}$, $\mathbf{b}_m^{(\ell)} \in \mathbb{R}^{U^{(m)}}$ and $\mathbf{h}_m^{(\ell)} \in \mathbb{R}^{U^{(m)}}$ for $m$-$th$ intermediate layer ($3 \leq m \leq C - 1$).

Since (3) must be calculated for any new subject in the testing phase, it is not computationally efficient. In other words, the transformed training data must be referenced by the current objective function for each new subject in the testing phase.

**Lemma 1.** *The equation (3) can be reformulated as follows where $\mathbf{G} \in \mathbb{R}^{T \times V_{new}}$ is the HA template:*

$$\min_{\mathbf{G}, \mathbf{R}^{(i)}, \theta^{(i)}} \sum_{i=1}^{S} \left\| \mathbf{G} - f_i\big(\mathbf{X}^{(i)}; \theta^{(i)}\big) \mathbf{R}^{(i)} \right\|_F^2 \quad \text{s.t.} \quad \mathbf{G}^\top \mathbf{G} = \mathbf{I}, \text{ where } \mathbf{G} = \frac{1}{S} \sum_{j=1}^{S} f_j\big(\mathbf{X}^{(j)}; \theta^{(j)}\big) \mathbf{R}^{(j)}. \tag{6}$$

*Proof.* In a nutshell, both (3) and (6) can be rewritten as $-S^2 \text{tr}\big(\mathbf{G}^\top \mathbf{G}\big) + \left( S \sum_{\ell=1}^{S} \text{tr}\left( \left( f_\ell\big(\mathbf{X}^{(\ell)}; \theta^{(\ell)}\big) \mathbf{R}^{(\ell)} \right)^\top f_\ell\big(\mathbf{X}^{(\ell)}; \theta^{(\ell)}\big) \mathbf{R}^{(\ell)} \right) \right)$. Please see supplementary materials for proof in details.

**Remark 1.** $\mathbf{G}$ is called DHA template, which can be used for functional alignment in the testing phase.

**Remark 2.** Same as previous approaches for HA problems [1–7], a DHA solution is not unique. If a DHA template $\mathbf{G}$ is calculated for a specific HA problem, then $\mathbf{QG}$ is another solution for that specific HA problem, where $\mathbf{Q} \in \mathbb{R}^{V_{new} \times V_{new}}$ can be any orthogonal matrix. Consequently, if two independent templates $\mathbf{G}_1$, $\mathbf{G}_2$ are trained for a specific dataset, the solutions can be mapped to each other by calculating $\left\| \mathbf{G}_2 - \mathbf{QG}_1 \right\|$, where $\mathbf{Q}$ can be used as a coefficient for functional alignment in the first solution in order to compare its results to the second one. Indeed, $\mathbf{G}_1$ and $\mathbf{G}_2$ are located in different positions on the same contour line [5, 7].

### 3.1 Optimization

This section proposes an effective approach for optimizing the DHA objective function by using rank-$m$ SVD [10] and SGD [13]. This method seeks an optimum solution for the DHA objective function (6) by using two different steps, which iteratively work in unison. By considering fixed network parameters $(\theta^{(\ell)})$, a mini-batch of neural activities is firstly aligned through the deep network. Then, back-propagation algorithm [14] is used to update the network parameters. The main challenge for solving the DHA objective function is that we cannot seek a natural extension of the correlation object to more than two random variables. Consequently, functional alignments are stacked in a $S \times S$ matrix and maximize a certain matrix norm for that matrix [10, 12].

As the first step, we consider network parameters are in an optimum state. Therefore, the mappings $(\mathbf{R}^{(\ell)}, \ell = 1{:}S)$ and template $(\mathbf{G})$ must be calculated to solve the DHA problem. In order to scale DHA approach, this paper employs the rank-$m$ SVD [10] of the mapped neural activities as follows:

$$f_\ell\big(\mathbf{X}^{(\ell)};\theta^{(\ell)}\big) \overset{SVD}{=} \mathbf{\Omega}^{(\ell)} \mathbf{\Sigma}^{(\ell)} \big(\mathbf{\Psi}^{(\ell)}\big)^\top, \qquad \ell = 1{:}S \tag{7}$$

where $\mathbf{\Sigma}^{(\ell)} \in \mathbb{R}^{m \times m}$ denotes the diagonal matrix with $m$-largest singular values of the mapped feature $f_\ell\big(\mathbf{X}^{(\ell)};\theta^{(\ell)}\big)$, $\mathbf{\Omega}^{(\ell)} \in \mathbb{R}^{T \times m}$ and $\mathbf{\Psi}^{(\ell)} \in \mathbb{R}^{m \times V_{new}}$ are respectively the corresponding left and right singular vectors. Based on (7), the projection matrix for $\ell$-$th$ subject can be generated as follows: [10]

$$\mathbf{P}^{(\ell)} = f_\ell\big(\mathbf{X}^{(\ell)};\theta^{(\ell)}\big) \left( \big(f_\ell\big(\mathbf{X}^{(\ell)};\theta^{(\ell)}\big)\big)^\top f_\ell\big(\mathbf{X}^{(\ell)};\theta^{(\ell)}\big) + \epsilon\mathbf{I} \right)^{-1} \big(f_\ell\big(\mathbf{X}^{(\ell)};\theta^{(\ell)}\big)\big)^\top$$
$$= \mathbf{\Omega}^{(\ell)}\big(\mathbf{\Sigma}^{(\ell)}\big)^\top \Big(\mathbf{\Sigma}^{(\ell)}\big(\mathbf{\Sigma}^{(\ell)}\big)^\top + \epsilon\mathbf{I}\Big)^{-1} \mathbf{\Sigma}^{(\ell)}\big(\mathbf{\Omega}^{(\ell)}\big)^\top = \mathbf{\Omega}^{(\ell)}\mathbf{D}^{(\ell)}\big(\mathbf{\Omega}^{(\ell)}\mathbf{D}^{(\ell)}\big)^\top, \tag{8}$$

where $\mathbf{P}^{(\ell)} \in \mathbb{R}^{T \times T}$ is symmetric and idempotent [10, 12], and diagonal matrix $\mathbf{D}^{(\ell)} \in \mathbb{R}^{m \times m}$ is

$$\mathbf{D}^{(\ell)}\big(\mathbf{D}^{(\ell)}\big)^\top = \big(\mathbf{\Sigma}^{(\ell)}\big)^\top \Big(\mathbf{\Sigma}^{(\ell)}\big(\mathbf{\Sigma}^{(\ell)}\big)^\top + \epsilon\mathbf{I}\Big)^{-1} \mathbf{\Sigma}^{(\ell)}. \tag{9}$$

Further, the sum of projection matrices can be defined as follows, where $\widetilde{\mathbf{A}}\widetilde{\mathbf{A}}^\top$ is the Cholesky decomposition [10] of $\mathbf{A}$:

$$\mathbf{A} = \sum_{i=1}^{S} \mathbf{P}^{(i)} = \widetilde{\mathbf{A}}\widetilde{\mathbf{A}}^\top, \quad \text{where} \quad \widetilde{\mathbf{A}} \in \mathbb{R}^{T \times mS} = \big[\mathbf{\Omega}^{(1)}\mathbf{D}^{(1)} \ldots \mathbf{\Omega}^{(S)}\mathbf{D}^{(S)}\big]. \tag{10}$$

**Lemma 2.** *Based on* (10)*, the objective function of DHA* (6) *can be rewritten as follows:*

$$\min_{\mathbf{G},\mathbf{R}^{(i)},\theta^{(i)}} \sum_{i=1}^{S} \left\| \mathbf{G} - f_i\big(\mathbf{X}^{(i)};\theta^{(i)}\big)\mathbf{R}^{(i)} \right\| \equiv \max_{\mathbf{G}} \Big(\text{tr}\big(\mathbf{G}^\top\mathbf{A}\mathbf{G}\big)\Big). \tag{11}$$

*Proof.* Since $\mathbf{P}^{(\ell)}$ is idempotent, the trace form of (6) can be reformulated as maximizing the sum of projections. Please see the supplementary materials for proof in details.

Based on Lemma 2, the first optimization step of DHA problem can be expressed as eigendecomposition of $\mathbf{A}\mathbf{G} = \mathbf{G}\mathbf{\Lambda}$, where $\Lambda = \{\lambda_1 \ldots \lambda_T\}$ and $\mathbf{G}$ respectively denote the eigenvalues and eigenvectors of $\mathbf{A}$. Further, the matrix $\mathbf{G}$ that we are interested in finding, can be calculated by the left singular vectors of $\widetilde{\mathbf{A}} = \mathbf{G}\widetilde{\mathbf{\Sigma}}\widetilde{\mathbf{\Psi}}^\top$, where $\mathbf{G}^\top\mathbf{G} = \mathbf{I}$ [10]. This paper utilizes Incremental SVD [15] for calculating these left singular vectors. Further, DHA mapping for $\ell$-$th$ subject is denoted as follows:

$$\mathbf{R}^{(\ell)} = \left( \big(f_\ell\big(\mathbf{X}^{(\ell)};\theta^{(\ell)}\big)\big)^\top f_\ell\big(\mathbf{X}^{(\ell)};\theta^{(\ell)}\big) + \epsilon\mathbf{I} \right)^{-1} \big(f_\ell\big(\mathbf{X}^{(\ell)};\theta^{(\ell)}\big)\big)^\top \mathbf{G}. \tag{12}$$

**Lemma 3.** *In order to update network parameters as the second step, the derivative of* $\mathbf{Z} = \sum_{\ell=1}^{T} \lambda_\ell$*, which is the sum of eigenvalues of* $\mathbf{A}$*, over the mapped neural activities of* $\ell$-$th$ *subject is defined as follows:*

$$\frac{\partial \mathbf{Z}}{\partial f_\ell\big(\mathbf{X}^{(\ell)};\theta^{(\ell)}\big)} = 2\mathbf{R}^{(\ell)}\mathbf{G}^\top - 2\mathbf{R}^{(\ell)}\big(\mathbf{R}^{(\ell)}\big)^\top \big(f_\ell\big(\mathbf{X}^{(\ell)};\theta^{(\ell)}\big)\big)^\top. \tag{13}$$

*Proof.* This derivative can be solved by using the chain and product rules in the matrix derivative as well as considering $\partial\mathbf{Z}/\partial\mathbf{A} = \mathbf{G}\mathbf{G}^\top$ [12]. Please see the supplementary materials for proof in details.

---

**Algorithm 1** Deep Hyperalignment (DHA)

---

**Input:** Data $\mathbf{X}^{(i)}$, $i = 1{:}S$, Regularized parameter $\epsilon$, Number of layers $C$, Number of units $U^{(m)}$ for $m = 2{:}C$, HA template $\widehat{\mathbf{G}}$ for testing phase (default $\emptyset$), Learning rate $\eta$ (default $10^{-4}$ [13]).
**Output:** DHA mappings $\mathbf{R}^{(\ell)}$ and parameters $\theta^{(\ell)}$, HA template $\mathbf{G}$ just from training phase
**Method:**
01. Initialize iteration counter: $m \leftarrow 1$ and $\theta^{(\ell)} \sim \mathcal{N}(0, 1)$ for $\ell = 1{:}S$.
02. Construct $f_\ell\big(\mathbf{X}^{(\ell)};\theta^{(\ell)}\big)$ based on (4) and (5) by using $\theta^{(\ell)}$, $C$, $U^{(m)}$ for $\ell = 1{:}S$.
03. **IF** ($\widehat{\mathbf{G}} \neq \emptyset$) **THEN**   % *The first step of DHA: fixed $\theta^{(\ell)}$ and calculating $\mathbf{G}$ and $\mathbf{R}^{(\ell)}$* ↓
04.    Generate $\widetilde{\mathbf{A}}$ by using (8) and (10).
05.    Calculate $\mathbf{G}$ by applying Incremental SVD [15] to $\widetilde{\mathbf{A}} = \mathbf{G}\widetilde{\mathbf{\Sigma}}\widetilde{\mathbf{\Psi}}^{\top}$.
06. **ELSE**
07.    $\mathbf{G} = \widehat{\mathbf{G}}$.
08. **END IF**
09. Calculate mappings $\mathbf{R}^{(\ell)}$, $\ell = 1{:}S$ by using (12).
10. Estimate error of iteration $\gamma_m = \sum_{i=1}^{S} \sum_{j=i+1}^{S} \left\| f_i\big(\mathbf{X}^{(i)};\theta^{(i)}\big)\mathbf{R}^{(i)} - f_j\big(\mathbf{X}^{(j)};\theta^{(j)}\big)\mathbf{R}^{(j)} \right\|_F^2$.
11. **IF** $\big((m > 3)$ and $(\gamma_m \geq \gamma_{m-1} \geq \gamma_{m-2})\big)$ **THEN**   % *This is the finishing condition.*
12.    **Return** calculated $\mathbf{G}$, $\mathbf{R}^{(\ell)}$, $\theta^{(\ell)}(\ell = 1{:}S)$ related to $(m{-}2)$-*th* iteration.
13. **END IF**   % *The second step of DHA: fixed $\mathbf{G}$ and $\mathbf{R}^{(\ell)}$ and updating $\theta^{(\ell)}$* ↓
14. $\nabla\theta^{(\ell)} \leftarrow \text{backprop}\Big(\partial\mathbf{Z}/\partial f_\ell(\mathbf{x}^{(\ell)};\theta^{(\ell)}), \theta^{(\ell)}\Big)$ by using (13) for $\ell = 1{:}S$.
15. Update $\theta^{(\ell)} \leftarrow \theta^{(\ell)} - \eta\nabla\theta^{(\ell)}$ for $\ell = 1{:}S$ and then $m \leftarrow m + 1$
16. **SAVE** all DHA parameters related to this iteration and **GO TO** Line *02*.

---

Algorithm 1 illustrates the DHA method for both training and testing phases. As depicted in this algorithm, (12) is just needed as the first step in the testing phase because the DHA template $\mathbf{G}$ is calculated for this phase based on the training samples (please see Lemma 1). As the second step in the DHA method, the networks' parameters ($\theta^{(\ell)}$) must be updated. This paper employs the back-propagation algorithm ($backprop()$ function) [14] as well as Lemma 3 for this step. In addition, finishing condition is defined by tackling errors in last three iterations, i.e. the average of the difference between each pair correlations of aligned functional activities across subjects ($\gamma_m$ for last three iterations). In other words, DHA will be finished if the error rates in the last three iterations are going to be worst. Further, a structure (nonlinear function for componentwise, and numbers of layers and units) for the deep network can be selected based on the optimum-state error ($\gamma_{opt}$) generated by training samples across different structures (see Experiment Schemes in the supplementary materials).

In summary, this paper proposes DHA as a flexible deep kernel approach to improve the performance of functional alignment in fMRI analysis. In order to seek an efficient functional alignment, DHA uses a deep network (*multiple stacked layers of nonlinear transformation*) for mapping fMRI responses of each subject to an embedded space ($f_\ell : \mathbb{R}^{T \times V} \to \mathbb{R}^{T \times V_{new}}, \ell = 1{:}S$). Unlike previous methods that use a restricted fixed kernel function, mapping functions in DHA are flexible across subjects because they employ multi-layer neural networks, which can implement any nonlinear function [12]. Therefore, DHA does not suffer from disadvantages of the previous kernel approach. In order to deal with high-dimensionality (broad ROI), DHA can also apply an optional feature selection by considering $V_{new} < V$ for constructing the deep networks. The performance of the optional feature selection will be analyzed in Section 4. Finally, DHA can be scaled across a large number of subjects by using the proposed optimization algorithm, i.e. rank-$m$ SVD, regularization, and mini-batch SGD.

## 4   Experiments

The empirical studies are reported in this section. Like previous studies [1–7, 9], this paper employs the $\nu$-SVM algorithms [16] for generating the classification model. Indeed, we use the binary $\nu$-SVM for datasets with just two categories of stimuli and multi-label $\nu$-SVM [3, 16] as the multi-class approach. All datasets are separately preprocessed by FSL 5.0.9 (`https://fsl.fmrib.ox.ac.uk`), i.e. slice timing, anatomical alignment, normalization, smoothing. Regions of Interests (ROI) are also denoted by employing the main reference of each dataset. In addition, leave-one-subject-out

Table 1: Accuracy of HA methods in post-alignment classification by using simple task datasets

| ↓Algorithms, Datasets→ | DS005 | DS105 | DS107 | DS116 | DS117 |
|---|---|---|---|---|---|
| $\nu$-SVM [17] | 71.65±0.97 | 22.89±1.02 | 38.84±0.82 | 67.26±1.99 | 73.32±1.67 |
| HA [1] | 81.27±0.59 | 30.03±0.87 | 43.01±0.56 | 74.23±1.40 | 77.93±0.29 |
| RHA [2] | 83.06±0.36 | 32.62±0.52 | 46.82±0.37 | 78.71±0.76 | 84.22±0.44 |
| KHA [3] | 85.29±0.49 | 37.14±0.36 | 52.69±0.69 | 78.03±0.89 | 83.32±0.41 |
| SVD-HA [4] | 90.82±1.23 | 40.21±0.83 | 59.54±0.99 | 81.56±0.54 | 95.62±0.83 |
| SRM [5] | 91.26±0.34 | 48.77±0.94 | 64.11±0.37 | 83.31±0.73 | 95.01±0.64 |
| SL [9] | 90.21±0.61 | 49.86±0.4 | 64.07±0.98 | 82.32±0.28 | 94.96±0.24 |
| CAE [6] | 94.25±0.76 | 54.52±0.80 | 72.16±0.43 | **91.49±0.67** | 95.92±0.67 |
| DHA | **97.92±0.82** | **60.39±0.68** | **73.05±0.63** | 90.28±0.71 | **97.99±0.94** |

Table 2: Area under the ROC curve (AUC) of different HA methods in post-alignment classification by using simple task datasets

| ↓Algorithms, Datasets→ | DS005 | DS105 | DS107 | DS116 | DS117 |
|---|---|---|---|---|---|
| $\nu$-SVM [17] | 68.37±1.01 | 21.76±0.91 | 36.84±1.45 | 62.49±1.34 | 70.17±0.59 |
| HA [1] | 70.32±0.92 | 28.91±1.03 | 40.21±0.33 | 70.67±0.97 | 76.14±0.49 |
| RHA [2] | 82.22±0.42 | 30.35±0.39 | 43.63±0.61 | 76.34±0.45 | 81.54±0.92 |
| KHA [3] | 80.91±0.21 | 36.23±0.57 | 50.41±0.92 | 75.28±0.94 | 80.92±0.28 |
| SVD-HA [4] | 88.54±0.71 | 37.61±0.62 | 57.54±0.31 | 78.66±0.82 | 92.14±0.42 |
| SRM [5] | 90.23±0.74 | 44.48±0.75 | 62.41±0.72 | 79.20±0.98 | 93.65±0.93 |
| SL [9] | 89.79±0.25 | 47.32±0.92 | 61.84±0.32 | 80.63±0.81 | 93.26±0.72 |
| CAE [6] | 91.24±0.61 | 52.16±0.63 | **72.33±0.79** | 87.53±0.72 | 91.49±0.33 |
| DHA | **96.91±0.82** | **59.57±0.32** | 70.23±0.92 | **89.93±0.24** | **96.13±0.32** |

cross-validation is utilized for partitioning datasets to the training set and testing set. Different HA methods are employed for functional aligning and then the mapped neural activities are used to generate the classification model. The performance of the proposed method is compared with the $\nu$-SVM algorithm as the baseline, where the features are used after anatomical alignment without applying any hyperalignment mapping. Further, performances of the standard HA [1], RHA [2], KHA [3], SVDHA [4], SRM [5], and SL [9] are reported as state-of-the-arts HA methods. In this paper, the results of HA algorithm is generated by employing Generalized CCA proposed in [10]. In addition, regularized parameters $(\alpha, \beta)$ in RHA are optimally assigned based on [2]. Further, KHA algorithm is used by the Gaussian kernel, which is evaluated as the best kernel in the original paper [3]. As another deep-learning-based alternative for functional alignment, the performance of CAE [6] is also compared with the proposed method. Like the original paper [6], this paper employs $k_1 = k_3 = \{5, 10, 15, 20, 25\}$, $\rho = \{0.1, 0.25, 0.5, 0.75, 0.9\}$, $\lambda = \{0.1, 1, 5, 10\}$. Then, aligned neural activities (by using CAE) are applied to the classification algorithm same as other HA techniques. This paper follows the CAE setup to set the same settings in the proposed method. Consequently, three hidden layers ($C = 5$) and the regularized parameters $\epsilon = \{10^{-4}, 10^{-6}, 10^{-8}\}$ are employed in the DHA method. In addition, the number of units in the intermediate layers are considered $U^{(m)} = KV$, where $m = 2{:}C\text{-}1$, $C$ is the number of layers, $V$ denotes the number of voxels and $K$ is the number of stimulus categories in each dataset[1]. Further, three distinctive activation functions are employed, i.e. Sigmoid ($g(\mathbf{x}) = 1/1 + \exp(-\mathbf{x})$), Hyperbolic ($g(\mathbf{x}) = \tanh(\mathbf{x})$), and Rectified Linear Unit or ReLU ($g(\mathbf{x}) = \ln(1 + \exp(\mathbf{x}))$). In this paper, the optimum parameters for DHA and CAE methods are reported for each dataset. Moreover, all algorithms are implemented by Python 3 on a PC with certain specifications[2] by authors in order to generate experimental results. Experiment schemes are also described in supplementary materials.

## 4.1 Simple Tasks Analysis

This paper utilizes 5 datasets, shared by Open fMRI (`https://openfmri.org`), for running empirical studies of this section. Further, numbers of original and aligned features are considered

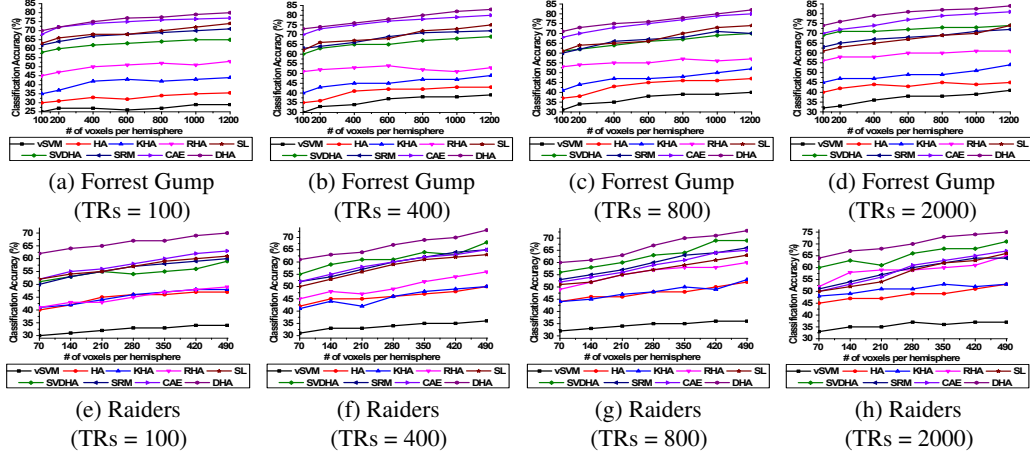

Figure 1: Comparison of different HA algorithms on complex task datasets by using ranked voxels.

equal ($V = V_{new}$) for all HA methods. As the first dataset, 'Mixed-gambles task' (DS005) includes $S = 48$ subjects. It also contains $K = 2$ categories of risk tasks in the human brain, where the chance of selection is $50/50$. In this dataset, the best results for CAE is generated by following parameters $k_1 = k_3 = 20, \rho = 0.75, \lambda = 1$ and for DHA by using $\epsilon = 10^{-8}$ and Hyperbolic function. In addition, ROI is defined based on the original paper [17]. As the second dataset, 'Visual Object Recognition' (DS105) includes $S = 71$ subjects. It also contains $K = 8$ categories of visual stimuli, i.e. gray-scale images of faces, houses, cats, bottles, scissors, shoes, chairs, and scrambles (nonsense patterns). In this dataset, the best results for CAE is generated by following parameters $k_1 = k_3 = 25, \rho = 0.9, \lambda = 5$ and for DHA by using $\epsilon = 10^{-6}$ and Sigmoid function. Please see [1, 7] for more information. As the third dataset, 'Word and Object Processing' (DS107) includes $S = 98$ subjects. It contains $K = 4$ categories of visual stimuli, i.e. words, objects, scrambles, consonants. In this dataset, the best results for CAE is generated by following parameters $k_1 = k_3 = 10, \rho = 0.5, \lambda = 10$ and for DHA by using $\epsilon = 10^{-6}$ and ReLU function. Please see [18] for more information. As the fourth dataset, 'Multi-subject, multi-modal human neuroimaging dataset' (DS117) includes MEG and fMRI images for $S = 171$ subjects. This paper just uses the fMRI images of this dataset. It also contains $K = 2$ categories of visual stimuli, i.e. human faces, and scrambles. In this dataset, the best results for CAE is generated by following parameters $k_1 = k_3 = 20, \rho = 0.9, \lambda = 5$ and for DHA by using $\epsilon = 10^{-8}$ and Sigmoid function. Please see [19] for more information. The responses of voxels in the Ventral Cortex are analyzed for these three datasets (DS105, DS107, DS117). As the last dataset, 'Auditory and Visual Oddball EEG-fMRI' (DS116) includes EEG signals and fMRI images for $S = 102$ subjects. This paper only employs the fMRI images of this dataset. It contains $K = 2$ categories of audio and visual stimuli, including oddball tasks. In this dataset, the best results for CAE is generated by following parameters $k_1 = k_3 = 10, \rho = 0.75, \lambda = 1$ and for DHA by using $\epsilon = 10^{-4}$ and ReLU function. In addition, ROI is defined based on the original paper [20]. This paper also provides the technical information of the employed datasets in the supplementary materials. Table 1 and 2 respectively demonstrate the classification Accuracy and Area Under the ROC Curve (AUC) in percentage (%) for the predictors. As these tables demonstrate, the performances of classification analysis without HA method are significantly low. Further, the proposed algorithm has generated better performance in comparison with other methods because it provided a better embedded space in order to align neural activities.

## 4.2 Complex Tasks Analysis

This section uses two fMRI datasets, which are related to watching movies. The numbers of original and aligned features are considered equal ($V = V_{new}$) for all HA methods. As the first dataset, 'A high-resolution 7-Tesla fMRI dataset from complex natural stimulation with an audio movie' (DS113) includes the fMRI data of $S = 18$ subjects, who watched 'Forrest Gump (1994)' movie during the experiment. This dataset provided by Open fMRI. In this dataset, the best results for CAE is generated by following parameters $k_1 = k_3 = 25, \rho = 0.9, \lambda = 10$ and for DHA by using $\epsilon = 10^{-8}$ and Sigmoid function. Please see [7] for more information. As the second dataset, $S = 10$ subjects watched 'Raiders of the Lost Ark (1981)', where whole brain volumes are 48. In this dataset, the best results for CAE is generated by following parameters $k_1 = k_3 = 15, \rho = 0.75, \lambda = 1$ and for DHA

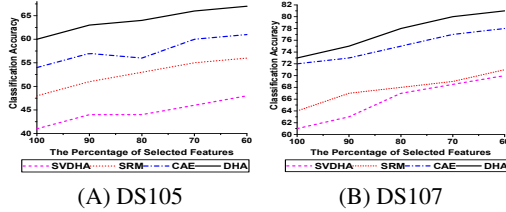
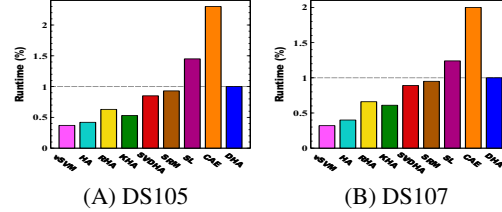

| (A) DS105 | (B) DS107 | (A) DS105 | (B) DS107 |
|---|---|---|---|

Figure 2: Classification by using feature selection.　　　Figure 3: Runtime Analysis

by using $\epsilon = 10^{-4}$ and Sigmoid function. Please see [3-5] for more information. In these two datasets, the ROI is defined in the ventral temporal cortex (VT). Figure 1 depicts the generated results, where the voxels in ROI are ranked by the method proposed in [1] based on their neurological priorities same as previous studies [1, 4, 7, 9]. Then, the experiments are repeated by using the different number of ranked voxels per hemisphere, i.e. in Forrest: $[100, 200, 400, 600, 800, 1000, 1200]$, and in Raiders: $[70, 140, 210, 280, 350, 420, 490]$. In addition, the empirical studies are reported by using the first $TRs = [100, 400, 800, 2000]$ in both datasets. Figure 1 shows that the DHA achieves superior performance to other HA algorithms.

### 4.3　Classification analysis by using feature selection

In this section, the effect of features selection ($V_{new} < V$) on the performance of classification methods will be discussed by using DS105 and DS107 datasets. Here, the performance of the proposed method is compared with SVDHA [4], SRM [5], and CAE [6] as the state-of-the-art HA techniques, which can apply feature selection before generating a classification model. Here, multi-label $\nu$-SVM [16] is used for generating the classification models after each of the mentioned methods applied on preprocessed fMRI images for functional alignment. In addition, the setup of this experiment is same as the previous sections (cross-validation, the best parameters, etc.). Figure 2 illustrates the performance of different methods by employing $100\%$ to $60\%$ of features. As depicted in this figure, the proposed method has generated better performance in comparison with other methods because it provides better feature representation in comparison with other techniques.

### 4.4　Runtime Analysis

In this section, the runtime of the proposed method is compared with the previous HA methods by using DS105 and DS107 datasets. As mentioned before, all of the results in this experiment are generated by a PC with certain specifications. Figure 3 illustrates the runtime of the mentioned methods, where runtime of other methods are scaled based on the DHA (runtime of the proposed method is considered as the unit). As depicted in this figure, CAE generated the worse runtime because it concurrently employs modified versions of SRM and SL for functional alignment. Further, SL also includes high time complexity because of the ensemble approach. By considering the performance of the proposed method in the previous sections, it generates acceptable runtime. As mentioned before, the proposed method employs rank-$m$ SVD [10] as well as Incremental SVD [15], which can significantly reduce the time complexity of the optimization procedure [10, 12].

## 5　Conclusion

This paper extended a deep approach for hyperalignment methods in order to provide accurate functional alignment in multi-subject fMRI analysis. Deep Hyperalignment (DHA) can handle fMRI datasets with nonlinearity, high-dimensionality (broad ROI), and a large number of subjects. We have also illustrated how DHA can be used for post-alignment classification. DHA is parametric and uses rank-$m$ SVD and stochastic gradient descent for optimization. Therefore, DHA generates low-runtime on large datasets, and DHA does not require the training data when the functional alignment is computed for a new subject. Further, DHA is not limited by a restricted fixed representational space because the kernel in DHA is a multi-layer neural network, which can separately implement any nonlinear function for each subject to transfer the brain activities to a common space. Experimental studies on multi-subject fMRI analysis confirm that the DHA method achieves superior performance to other state-of-the-art HA algorithms. In the future, we will plan to employ DHA for improving the performance of other techniques in fMRI analysis, e.g. Representational Similarity Analysis (RSA).

## Acknowledgments

This work was supported in part by the National Natural Science Foundation of China (61422204, 61473149, and 61732006), and NUAA Fundamental Research Funds (NE2013105).

## Footnotes

[1]Although we can use any settings for DHA, we empirically figured out this setting is acceptable to seek an optimum solution. Indeed, we followed CAE setup in the network structure but used the number of categories ($K$) rather than a series of parameters. In the current format of DHA, we just need to set the regularized constant and the nonlinear activation function, while a wide range of parameters must be set in the CAE.

[2]DEL, CPU = Intel Xeon E5-2630 v3 (8×2.4 GHz), RAM = 64GB, GPU = GeForce GTX TITAN X (12GB memory), OS = Ubuntu 16.04.3 LTS, Python = 3.6.2, Pip = 9.0.1, Numpy = 1.13.1, Scipy = 0.19.1, Scikit-Learn = 0.18.2, Theano = 0.9.0.

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
