[Supplementary Material 1]

# Deep Hyperalignment

## Supplementary Materials

**Muhammad Yousefnezhad & Daoqiang Zhang**
College of Computer Science and Technology
Nanjing University of Aeronautics and Astronautics
{myousefnezhad,dqzhang}@nuaa.edu.cn

## S.1 Notations

Table S1 illustrates all variables and functions that are employed in this paper.

## S.2 Visualizing DHA Common Space (G)

This section presents a visualized example of the DHA common space ($\mathbf{G}$). Here, we have firstly applied DHA to DS105 dataset and generated the common space by considering $V = V_{new}$ same as SRM [5] and CAE [6] methods. As depicted in Fig. S1, the average of neural activities in each voxel is calculated and visualized across categories of visual stimuli.

## S.3 Technical Information

Figure S2 illustrates the graphical abstract of DHA, where the hyperalignment applied to a problem with 3-dimensional space for mapping the neural activities to a common space with 2-dimensions. In addition, Table S2 demonstrates the technical information of the employed datasets in this paper.

## S.4 Overview of Hyperalignment methods

Preprocessed fMRI time series collected for $S$ subjects can be defined by $\mathbf{X}^{(i)} = \left\{ x_{mn}^{(i)} \right\} \in \mathbb{R}^{T \times V}$, $i = 1{:}S$, $m = 1{:}T$, $n = 1{:}V$, where $T$ denotes the number of time points in unites of TRs (Time of Repetition), $V$ is the number of voxels, and $x_{mn}^{(i)} \in \mathbb{R}$ denotes the functional activity for the $i$-$th$ subject in the $m$-$th$ time point and the $n$-$th$ voxel. For simplicity assume that all data points are normalized by zero-mean and unit-variance, $\mathbf{X}^{(i)} \sim \mathcal{N}(0,1)$, $i = 1{:}S$. If the original data points are not normalized, we can consider this assumption as a preprocessing step. Since there are more voxels than TRs in most of the fMRI studies, $\mathbf{X}^{(i)}$ and the voxel correlation map $(\mathbf{X}^{(i)})^\top \mathbf{X}^{(j)}$ may not be full rank [1–4]. In training-set, time synchronized stimulus ensures temporal alignment, i.e. the $m$-$th$ time point for all of the subjects represents the same simulation [2, 3]. Indeed, the main goal of HA methods is aligning the columns of $\mathbf{X}^{(i)}$ across subjects [2]. In previous studies, Inter-Subject Correlation (ISC) was defined for functional alignment between two distinct subjects as follows, where $tr()$ is the trace function: [1–4]

$$\text{ISC}(\mathbf{X}^{(i)}, \mathbf{X}^{(j)}) = (1/V)\text{tr}\Big((\mathbf{X}^{(i)})^\top \mathbf{X}^{(j)}\Big). \tag{S.1}$$

By considering $\mathbf{X}^{(\ell)} \sim \mathcal{N}(0,1)$, $\ell = 1{:}S$ as column-wise standardized, ISC lies in $[-1, +1]$. Here, the large values illustrate better alignment [2, 4]. Based on (S.1), Hyperalignment (HA) problem can

be defined as follows: [1–3]

$$\max_{\mathbf{R}^{(i)},\mathbf{R}^{(j)}} \sum_{i=1}^{S} \sum_{j=i+1}^{S} \mathrm{ISC}(\mathbf{X}^{(i)}\mathbf{R}^{(i)},\mathbf{X}^{(j)}\mathbf{R}^{(j)}) \qquad \text{s.t.} \quad \left(\mathbf{R}^{(\ell)}\right)^{\top}\widetilde{\boldsymbol{\Phi}}^{(\ell)}\mathbf{R}^{(\ell)} = \mathbf{I}, \ell = 1{:}S, \quad \text{(S.2)}$$

where $\mathbf{I}$ is the identity matrix, $\mathbf{R}^{(\ell)} = \left\{ r_{mn}^{(\ell)} \right\} \in \mathbb{R}^{V \times V}$ denotes the solution for $\ell\text{-}th$ subject, and the matrices $\widetilde{\boldsymbol{\Phi}}^{(\ell)} \in \mathbb{R}^{V \times V}, \ell = 1{:}S$ are symmetric and positive definite. By considering $\widetilde{\boldsymbol{\Phi}}^{(\ell)} = \mathbf{I}$, (S.2) is equivalent to a multi-set orthogonal Procrustes problem, which is commonly used in share analysis. Furthermore, if $\widetilde{\boldsymbol{\Phi}}^{(\ell)} = (\mathbf{X}^{(\ell)})^{\top}\mathbf{X}^{(\ell)}$, then (S.2) denotes a form of multi-set Canonical Correlation Analysis (CCA) [3, 4] as follows:

$$\max_{\mathbf{R}^{(i)},\mathbf{R}^{(j)}} \sum_{i=1}^{S} \sum_{j=i+1}^{S} \mathrm{tr}\left((\mathbf{X}^{(i)}\mathbf{R}^{(i)})^{\top}\mathbf{X}^{(j)}\mathbf{R}^{(j)}\right) \quad \text{s.t.} \quad \left(\mathbf{X}^{(\ell)}\mathbf{R}^{(\ell)}\right)^{\top}\mathbf{X}^{(\ell)}\mathbf{R}^{(\ell)} = \mathbf{I}, \quad \ell = 1{:}S.$$

$$\text{(S.3)}$$

**Lemma S.1.** (S.3) *maximizes the Pearson correlation* $(cov(A,B)/\sigma_A\sigma_B)$ *between each pair of mapped features* $(\mathbf{X}^{(\ell)}\mathbf{R}^{(\ell)}$ *) across subjects.*

*Proof.* By considering $\widetilde{\boldsymbol{\Phi}}^{(\ell)} \in \mathbb{R}^{V \times V} = \mathbb{E}\left[(\mathbf{X}^{(\ell)})^{\top}\mathbf{X}^{(\ell)}\right] = (\mathbf{X}^{(\ell)})^{\top}\mathbf{X}^{(\ell)}$, the Pearson correlation on (S.3) can be defined as follows:

$$\max_{\mathbf{R}^{(i)},\mathbf{R}^{(j)}} \sum_{i=1}^{S} \sum_{j=i+1}^{S} \mathrm{tr}\left(\frac{(\mathbf{X}^{(i)}\mathbf{R}^{(i)})^{\top}\mathbf{X}^{(i)}\mathbf{R}^{(j)}}{\sqrt{((\mathbf{R}^{(i)})^{\top}\widetilde{\boldsymbol{\Phi}}^{(i)}\mathbf{R}^{(i)})}\sqrt{((\mathbf{R}^{(j)})^{\top}\widetilde{\boldsymbol{\Phi}}^{(j)}\mathbf{R}^{(j)})}}\right). \qquad \text{(S.4)}$$

Since $\left(\mathbf{R}^{(\ell)}\right)^{\top}\widetilde{\boldsymbol{\Phi}}^{(\ell)}\mathbf{R}^{(\ell)} = \mathbf{I}$, we approach from (S.4) to (S.3).

In fact, the constrains must be imposed in $\mathbf{R}^{(\ell)}$ to avoid overfitting [2]. In order to seek an optimum solution, solving (S.3) may not be the best approach because there is no scale to evaluate the distance between current result and the optimum (fully maximized) solution [2, 4, 7].

**Lemma S.2.** (S.3) *can be rewritten as following minimization problem:*

$$\min_{\mathbf{R}^{(i)},\mathbf{R}^{(j)}} \sum_{i=1}^{S} \sum_{j=i+1}^{S} \left\|\mathbf{X}^{(i)}\mathbf{R}^{(i)} - \mathbf{X}^{(j)}\mathbf{R}^{(j)}\right\|_{F}^{2}, \quad \text{s.t.} \quad \left(\mathbf{X}^{(\ell)}\mathbf{R}^{(\ell)}\right)^{\top}\mathbf{X}^{(\ell)}\mathbf{R}^{(\ell)} = \mathbf{I}, \quad \ell = 1{:}S,$$

$$\text{(S.5)}$$

*where* (S.5) *approaches zero for an optimum result.*
*Proof.*

$$\min_{\mathbf{R}^{(i)},\mathbf{R}^{(j)}} \sum_{i=1}^{S} \sum_{j=i+1}^{S} \left\|\mathbf{X}^{(i)}\mathbf{R}^{(i)} - \mathbf{X}^{(j)}\mathbf{R}^{(j)}\right\|_{F}^{2} = \min_{\mathbf{R}^{(i)},\mathbf{R}^{(j)}} \sum_{i=1}^{S} \sum_{j=i+1}^{S} \left(\mathrm{tr}\left((\mathbf{X}^{(i)}\mathbf{R}^{(i)})^{\top}\mathbf{X}^{(i)}\mathbf{R}^{(i)}\right) + \right.$$

$$\left. \mathrm{tr}\left((\mathbf{X}^{(j)}\mathbf{R}^{(j)})^{\top}\mathbf{X}^{(j)}\mathbf{R}^{(j)}\right) - 2\mathrm{tr}\left((\mathbf{X}^{(i)}\mathbf{R}^{(i)})^{\top}\mathbf{X}^{(j)}\mathbf{R}^{(j)}\right)\right).$$

Since $\left(\mathbf{X}^{(\ell)}\mathbf{R}^{(\ell)}\right)^{\top}\mathbf{X}^{(\ell)}\mathbf{R}^{(\ell)} = \mathbf{I}$, so we have:

$$\min_{\mathbf{R}^{(i)},\mathbf{R}^{(j)}} \sum_{i=1}^{S} \sum_{j=i+1}^{S} \left(2V_{new} - 2\mathrm{tr}\left((\mathbf{X}^{(i)}\mathbf{R}^{(i)})^{\top}\mathbf{X}^{(j)}\mathbf{R}^{(j)}\right)\right) \equiv \max_{\mathbf{R}^{(i)},\mathbf{R}^{(j)}} \sum_{i=1}^{S} \sum_{j=i+1}^{S} \mathrm{tr}\left((\mathbf{X}^{(i)}\mathbf{R}^{(i)})^{\top}\mathbf{X}^{(j)}\mathbf{R}^{(j)}\right).$$

Indeed, the main assumption in the original HA is that the $\mathbf{R}^{(\ell)}, \ell = 1{:}S$ are noisy 'rotations' of a common template [1, 9]. In order to extend HA into the real-world problems, (S.5) can be reformulated by using an embedded space as follows:

$$\min_{\mathbf{R}^{(i)},\mathbf{R}^{(j)}} \sum_{i=1}^{S} \sum_{j=i+1}^{S} \left\|f(\mathbf{X}^{(i)})\mathbf{R}^{(i)} - f(\mathbf{X}^{(j)})\mathbf{R}^{(j)}\right\|_{F}^{2}, \text{s.t.} \left(f(\mathbf{X}^{(\ell)})\mathbf{R}^{(\ell)}\right)^{\top} f(\mathbf{X}^{(\ell)})\mathbf{R}^{(\ell)} = \mathbf{I}, \ell = 1{:}S.$$

$$\text{(S.6)}$$

Here, if $f(\mathbf{x}) = \mathbf{x}$, then we recover the original HA. Further, if $f(\mathbf{x})$ denotes any classical fixed kernel function (e.g. Gaussian), then (S.6) is equivalent to Kernel Hyperalignment (KHA) [3]. In addition, if $f(\mathbf{x})$ illustrates an SVD-based feature selection, then (S.6) is called SVD Hyperalignment (SVDHA) [4]. Finally, this paper proposes Deep Hyperalignment (DHA), where $f_\ell(\mathbf{X}^{(\ell)};\theta^{(\ell)})$ is defined (rather than $f(\mathbf{x})$) as a parametric deep neural network across subjects, i.e. multiple stacked layers of nonlinear transformation for each subject. As another alternative, (S.6) can be optimized by using different approaches. For instance, Regularized Hyperalignment (RHA) used the Generalized CCA $\left(\min\limits_{\mathbf{G},\mathbf{R}^{(i)}} \sum_{i=1}^{S} \left\| \mathbf{G} - \mathbf{X}^{(i)}\mathbf{R}^{(i)} \right\|_F^2 \right)$ [2], Shared Response Model (SRM) employed Probabilistic CCA [5]. This paper also uses rank-m SVD [9] and Stochastic Gradient Descent (SGD) [12] for optimization.

## S.5 Equations

This section presents the equations of the original paper that we need them in order to proof Lemmas in the next section.

1. Objective function of DHA is defined as follows:

$$\min_{\substack{\theta^{(i)},\mathbf{R}^{(i)} \\ \theta^{(j)},\mathbf{R}^{(j)}}} \sum_{i=1}^{S} \sum_{j=i+1}^{S} \left\| f_i(\mathbf{X}^{(i)};\theta^{(i)})\mathbf{R}^{(i)} - f_j(\mathbf{X}^{(j)};\theta^{(j)})\mathbf{R}^{(j)} \right\|_F^2 \tag{3}$$

$$\text{s.t.} \quad \left(\mathbf{R}^{(\ell)}\right)^\top \left( \left(f_\ell(\mathbf{X}^{(\ell)};\theta^{(\ell)})\right)^\top f_\ell(\mathbf{X}^{(\ell)};\theta^{(\ell)}) + \epsilon\mathbf{I}\right)\mathbf{R}^{(\ell)} = \mathbf{I}, \quad \ell = 1{:}S.$$

2. Projection matrix for $\ell\text{-}th$ subject can be generated as follows: [10]

$$\mathbf{P}^{(\ell)} = f_\ell(\mathbf{X}^{(\ell)};\theta^{(\ell)}) \left( \left(f_\ell(\mathbf{X}^{(\ell)};\theta^{(\ell)})\right)^\top f_\ell(\mathbf{X}^{(\ell)};\theta^{(\ell)}) + \epsilon\mathbf{I}\right)^{-1} \left(f_\ell(\mathbf{X}^{(\ell)};\theta^{(\ell)})\right)^\top$$

$$= \mathbf{\Omega}^{(\ell)}\left(\mathbf{\Sigma}^{(\ell)}\right)^\top \left(\mathbf{\Sigma}^{(\ell)}\left(\mathbf{\Sigma}^{(\ell)}\right)^\top + \epsilon\mathbf{I}\right)^{-1} \mathbf{\Sigma}^{(\ell)}\left(\mathbf{\Omega}^{(\ell)}\right)^\top = \mathbf{\Omega}^{(\ell)}\mathbf{D}^{(\ell)}\left(\mathbf{\Omega}^{(\ell)}\mathbf{D}^{(\ell)}\right)^\top, \tag{8}$$

where diagonal matrix $\mathbf{D}^{(\ell)} \in \mathbb{R}^{m \times m}$ is

$$\mathbf{D}^{(\ell)}\left(\mathbf{D}^{(\ell)}\right)^\top = \left(\mathbf{\Sigma}^{(\ell)}\right)^\top \left(\mathbf{\Sigma}^{(\ell)}\left(\mathbf{\Sigma}^{(\ell)}\right)^\top + \epsilon\mathbf{I}\right)^{-1}\mathbf{\Sigma}^{(\ell)}. \tag{9}$$

3. The sum of projection can be defined as follows:

$$\mathbf{A} = \sum_{i=1}^{S} \mathbf{P}^{(i)} = \widetilde{\mathbf{A}}\left(\widetilde{\mathbf{A}}\right)^\top, \quad \text{where} \quad \widetilde{\mathbf{A}} \in \mathbb{R}^{T \times mS} = \left[\mathbf{\Omega}^{(1)}\mathbf{D}^{(1)} \dots \mathbf{\Omega}^{(S)}\mathbf{D}^{(S)}\right]. \tag{10}$$

4. DHA mapping for $\ell\text{-}th$ subject is denoted as follows:

$$\mathbf{R}^{(\ell)} = \left( \left(f_\ell(\mathbf{X}^{(\ell)};\theta^{(\ell)})\right)^\top f_\ell(\mathbf{X}^{(\ell)};\theta^{(\ell)}) + \epsilon\mathbf{I}\right)^{-1} \left(f_\ell(\mathbf{X}^{(\ell)};\theta^{(\ell)})\right)^\top \mathbf{G}. \tag{12}$$

## S.6 Proofs

**Lemma 1.** *The equation* (3) *can be reformulated as follows where* $\mathbf{G} \in \mathbb{R}^{T \times V_{new}}$ *is the HA template:*

$$\min_{\mathbf{G},\mathbf{R}^{(i)},\theta^{(i)}} \sum_{i=1}^{S} \left\| \mathbf{G} - f_i(\mathbf{X}^{(i)};\theta^{(i)})\mathbf{R}^{(i)} \right\|_F^2 \quad \text{s.t.} \quad \mathbf{G}^\top\mathbf{G} = \mathbf{I}, \text{ where } \mathbf{G} = \frac{1}{S}\sum_{j=1}^{S} f_j(\mathbf{X}^{(j)};\theta^{(j)})\mathbf{R}^{(j)}. \tag{6}$$

*Proof.* For simplicity, we consider that $f_\ell\big(\mathbf{X}^{(\ell)};\theta^{(\ell)}\big)=\mathbf{Y}^{(\ell)}$:

$$\min_{\substack{\theta^{(i)},\mathbf{R}^{(i)}\\ \theta^{(j)},\mathbf{R}^{(j)}}}\sum_{i=1}^{S}\sum_{j=i+1}^{S}\left\|f_i\big(\mathbf{X}^{(i)};\theta^{(i)}\big)\mathbf{R}^{(i)}-f_j\big(\mathbf{X}^{(j)};\theta^{(j)}\big)\mathbf{R}^{(j)}\right\|_F^2=\min_{\substack{\theta^{(i)},\mathbf{R}^{(i)}\\ \theta^{(j)},\mathbf{R}^{(j)}}}\sum_{i=1}^{S}\sum_{j=i+1}^{S}\left\|\mathbf{Y}^{(i)}\mathbf{R}^{(i)}-\mathbf{Y}^{(j)}\mathbf{R}^{(j)}\right\|_F^2$$

$$=\min_{\substack{\theta^{(i)},\mathbf{R}^{(i)}\\ \theta^{(j)},\mathbf{R}^{(j)}}}\sum_{i=1}^{S}\sum_{j=i+1}^{S}\left(\operatorname{tr}\Big(\big(\mathbf{Y}^{(i)}\mathbf{R}^{(i)}\big)^{\top}\mathbf{Y}^{(i)}\mathbf{R}^{(i)}\Big)+\operatorname{tr}\Big(\big(\mathbf{Y}^{(j)}\mathbf{R}^{(j)}\big)^{\top}\mathbf{Y}^{(j)}\mathbf{R}^{(j)}\Big)-2\operatorname{tr}\Big(\big(\mathbf{Y}^{(i)}\mathbf{R}^{(i)}\big)^{\top}\mathbf{Y}^{(j)}\mathbf{R}^{(j)}\Big)\right)$$

$$\equiv\min_{\substack{\theta^{(i)},\mathbf{R}^{(i)}\\ \theta^{(j)},\mathbf{R}^{(j)}\\ \mathbf{G}}}\frac{1}{2}\sum_{i=1}^{S}\left(S\operatorname{tr}\Big(\big(\mathbf{Y}^{(i)}\mathbf{R}^{(i)}\big)^{\top}\mathbf{Y}^{(i)}\mathbf{R}^{(i)}\Big)-2S\operatorname{tr}\Big(\big(\mathbf{Y}^{(i)}\mathbf{R}^{(i)}\big)^{\top}\mathbf{G}\Big)+\sum_{j=1}^{S}\Big(\operatorname{tr}\Big(\big(\mathbf{Y}^{(j)}\mathbf{R}^{(j)}\big)^{\top}\mathbf{Y}^{(j)}\mathbf{R}^{(j)}\Big)\Big)\right)$$

$$=\min_{\substack{\theta^{(i)},\mathbf{R}^{(i)}\\ \theta^{(j)},\mathbf{R}^{(j)}\\ \mathbf{G}}}\frac{1}{2}\left(\Big(S\sum_{i=1}^{S}\operatorname{tr}\Big(\big(\mathbf{Y}^{(i)}\mathbf{R}^{(i)}\big)^{\top}\mathbf{Y}^{(i)}\mathbf{R}^{(i)}\Big)\Big)-\Big(2S^2\operatorname{tr}(\mathbf{G}^{\top}\mathbf{G})\Big)+\Big(S\sum_{j=1}^{S}\operatorname{tr}\Big(\big(\mathbf{Y}^{(j)}\mathbf{R}^{(j)}\big)^{\top}\mathbf{Y}^{(j)}\mathbf{R}^{(j)}\Big)\Big)\right)$$

$$=\min_{\substack{\theta^{(i)},\mathbf{R}^{(i)}\\ \mathbf{G}}}\frac{1}{2}\left(\Big(2S\sum_{i=1}^{S}\operatorname{tr}\Big(\big(\mathbf{Y}^{(i)}\mathbf{R}^{(i)}\big)^{\top}\mathbf{Y}^{(i)}\mathbf{R}^{(i)}\Big)\Big)-\Big(2S^2\operatorname{tr}(\mathbf{G}^{\top}\mathbf{G})\Big)\right)$$

$$=\min_{\substack{\theta^{(i)},\mathbf{R}^{(i)}\\ \mathbf{G}}}\left(\Big(S\sum_{i=1}^{S}\operatorname{tr}\Big(\big(\mathbf{Y}^{(i)}\mathbf{R}^{(i)}\big)^{\top}\mathbf{Y}^{(i)}\mathbf{R}^{(i)}\Big)\Big)-\Big(S^2\operatorname{tr}(\mathbf{G}^{\top}\mathbf{G})\Big)\right)$$

$$=\min_{\substack{\theta^{(i)},\mathbf{R}^{(i)}\\ \mathbf{G}}}\left(\Big(S\sum_{i=1}^{S}\operatorname{tr}\Big(\big(\mathbf{Y}^{(i)}\mathbf{R}^{(i)}\big)^{\top}\mathbf{Y}^{(i)}\mathbf{R}^{(i)}\Big)\Big)-2\Big(S^2\operatorname{tr}(\mathbf{G}^{\top}\mathbf{G})\Big)+\Big(S^2\operatorname{tr}(\mathbf{G}^{\top}\mathbf{G})\Big)\right)$$

$$=\min_{\substack{\theta^{(i)},\mathbf{R}^{(i)}\\ \mathbf{G}}}S\sum_{i=1}^{S}\left(\operatorname{tr}\Big(\big(\mathbf{Y}^{(i)}\mathbf{R}^{(i)}\big)^{\top}\mathbf{Y}^{(i)}\mathbf{R}^{(i)}\Big)+\operatorname{tr}(\mathbf{G}^{\top}\mathbf{G})-2\operatorname{tr}\Big(\big(\mathbf{Y}^{(i)}\mathbf{R}^{(i)}\big)^{\top}\mathbf{G}\Big)\right)$$

$$\equiv\min_{\substack{\theta^{(i)},\mathbf{R}^{(i)}\\ \mathbf{G}}}\sum_{i=1}^{S}\|\mathbf{G}-\mathbf{Y}^{(i)}\mathbf{R}^{(i)}\|_F^2=\min_{\mathbf{G},\mathbf{R}^{(i)},\theta^{(i)}}\sum_{i=1}^{S}\left\|\mathbf{G}-f_i\big(\mathbf{X}^{(i)};\theta^{(i)}\big)\mathbf{R}^{(i)}\right\|_F^2\ \blacksquare$$

**Lemma 2.** *Based on* (10)*, the objective function of DHA* (6) *can be rewritten as follows:*

$$\min_{\mathbf{G},\mathbf{R}^{(i)},\theta^{(i)}}\sum_{i=1}^{S}\left\|\mathbf{G}-f_i\big(\mathbf{X}^{(i)};\theta^{(i)}\big)\mathbf{R}^{(i)}\right\|\equiv\max_{\mathbf{G}}\Big(\operatorname{tr}(\mathbf{G}^{\top}\mathbf{A}\mathbf{G})\Big). \tag{10}$$

*Proof.* For simplicity, we consider that $f_\ell\big(\mathbf{X}^{(\ell)};\theta^{(\ell)}\big)=\mathbf{Y}^{(\ell)}$. By considering (12), we have:

$$\min_{\mathbf{G},\theta^{(i)},\mathbf{R}^{(i)}}\sum_{i=1}^{S}\left\|\mathbf{G}-\mathbf{Y}^{(i)}\Big(\big(\mathbf{Y}^{(i)}\big)^{\top}\mathbf{Y}^{(i)}+\epsilon\mathbf{I}\Big)^{-1}\big(\mathbf{Y}^{(i)}\big)^{\top}\mathbf{G}\right\|_F^2$$

Based on (8), we have:

$$\min_{\mathbf{G}}\sum_{i=1}^{S}\|\mathbf{G}-\mathbf{P}^{(i)}\mathbf{G}\|_F^2=\min_{\mathbf{G}}\sum_{i=1}^{S}\|(\mathbf{I}-\mathbf{P}^{(i)})\mathbf{G}\|_F^2=\min_{\mathbf{G}}\sum_{i=1}^{S}\operatorname{tr}\Big(\big((\mathbf{I}-\mathbf{P}^{(i)})\mathbf{G}\big)^{\top}(\mathbf{I}-\mathbf{P}^{(i)})\mathbf{G}\Big)$$

$$=\min_{\mathbf{G}}\sum_{i=1}^{S}\operatorname{tr}\Big(\mathbf{G}^{\top}(\mathbf{I}-\mathbf{P}^{(i)})^{\top}(\mathbf{I}-\mathbf{P}^{(i)})\mathbf{G}\Big)=\min_{\mathbf{G}}\sum_{i=1}^{S}\operatorname{tr}\Big(\mathbf{G}^{\top}(\mathbf{I}-\mathbf{P}^{(i)})^2\mathbf{G}\Big)$$

Since $\mathbf{P}^{(i)}$ is idempotent ($(\mathbf{P}^{(i)})^2 = \mathbf{P}^{(i)}$) [10, 11], we have:

$$\min_{\mathbf{G}} \sum_{i=1}^{S} \mathrm{tr}\Big( \mathbf{G}^{\top}(\mathbf{I} - \mathbf{P}^{(i)})^2 \mathbf{G} \Big) = \min_{\mathbf{G}} \sum_{i=1}^{S} \mathrm{tr}\Big( \mathbf{G}^{\top}\big(\mathbf{I}^2 + (\mathbf{P}^{(i)})^2 - 2\mathbf{I}\mathbf{P}^{(i)}\big)\mathbf{G} \Big)$$

$$= \min_{\mathbf{G}} \sum_{i=1}^{S} \mathrm{tr}\Big( \mathbf{G}^{\top}\big(\mathbf{I}^2 + \mathbf{P}^{(i)} - 2\mathbf{P}^{(i)}\big)\mathbf{G} \Big) = \min_{\mathbf{G}} \sum_{i=1}^{S} \mathrm{tr}\Big( \mathbf{G}^{\top}(\mathbf{I} - \mathbf{P}^{(i)})\mathbf{G} \Big)$$

$$= \min_{\mathbf{G}} \sum_{i=1}^{S} \mathrm{tr}\Big( \mathbf{G}^{\top}\mathbf{I}\mathbf{G} - \mathbf{G}^{\top}\mathbf{P}^{(i)}\mathbf{G} \Big) = \min_{\mathbf{G}} \sum_{i=1}^{S} \Big( \mathrm{tr}(\mathbf{I}) - \mathrm{tr}(\mathbf{G}^{\top}\mathbf{P}^{(i)}\mathbf{G}) \Big)$$

$$= \min_{\mathbf{G}} \Big( SV - \mathrm{tr}\Big( \mathbf{G}^{\top}\big(\sum_{i=1}^{S}\mathbf{P}^{(i)}\big)\mathbf{G}\Big) \Big)$$

Based on (10), we have:

$$= \min_{\mathbf{G}} \Big( SV - \mathrm{tr}\Big(\mathbf{G}^{\top}\mathbf{A}\mathbf{G}\Big) \Big) \equiv \max_{\mathbf{G}} \Big(\mathrm{tr}(\mathbf{G}^{\top}\mathbf{A}\mathbf{G})\Big) \quad \blacksquare$$

**Lemma 3.** *In order to update network parameters as the second step, the derivative of* $\mathbf{Z} = \sum_{\ell=1}^{T} \lambda_{\ell}$, *which is the sum of eigenvalues of* $\mathbf{A}$, *over the mapped neural activities of* $\ell$-*th subject is defined as follows:*

$$\frac{\partial \mathbf{Z}}{\partial f_{\ell}\big(\mathbf{X}^{(\ell)};\theta^{(\ell)}\big)} = 2\mathbf{R}^{(\ell)}\mathbf{G}^{\top} - 2\mathbf{R}^{(\ell)}\big(\mathbf{R}^{(\ell)}\big)^{\top}\Big(f_{\ell}(\mathbf{X}^{(\ell)};\theta^{(\ell)})\Big)^{\top} \tag{13}$$

*Proof.* Based on the parameters (regularization, template, mappings, etc.) in DHA, we present a modified version of the Appendix A proof proposed in [11]. For simplicity, we define $f_{\ell}\big(\mathbf{X}^{(\ell)};\theta^{(\ell)}\big) = \mathbf{Y}^{(\ell)}$ and the covariance matrix for $\ell$-*th* subject as follows:

$$\widetilde{\boldsymbol{\Phi}}^{(\ell)} = \Big(\big(\mathbf{Y}^{(\ell)}\big)^{\top}\mathbf{Y}^{(\ell)} + \epsilon\mathbf{I}\Big) \tag{S.7}$$

Further, the inverse of the covariance matrix is denoted for $\ell$-*th* subject as follows:

$$\boldsymbol{\Phi}^{(\ell)} = \big(\widetilde{\boldsymbol{\Phi}}^{(\ell)}\big)^{-1} = \Big(\big(\mathbf{Y}^{(\ell)}\big)^{\top}\mathbf{Y}^{(\ell)} + \epsilon\mathbf{I}\Big)^{-1} \tag{S.8}$$

where this matrix is symmetric ($\boldsymbol{\Phi}^{(\ell)} = \big(\boldsymbol{\Phi}^{(\ell)}\big)^{\top}$). Based on (S.8), the projection (8) can be rewritten as follows:

$$\mathbf{P}^{(\ell)} = \mathbf{Y}^{(\ell)}\boldsymbol{\Phi}^{(\ell)}\big(\mathbf{Y}^{(\ell)}\big)^{\top} \tag{S.9}$$

Further, the final mappings (12) can be reformulated as follows:

$$\mathbf{R}^{(\ell)} = \boldsymbol{\Phi}^{(\ell)}\big(\mathbf{Y}^{(\ell)}\big)^{\top}\mathbf{G} \tag{S.10}$$

Since $\partial\mathbf{Z}/\partial\mathbf{A} = \mathbf{G}\mathbf{G}^{\top}$ [14], we reformulate the left side of (13) based on the chain rule as follows:

$$\frac{\partial \mathbf{Z}}{\partial \mathbf{Y}^{(\ell)}_{\alpha\beta}} = \sum_{\mu,\tau=1}^{V_{new}} \frac{\partial \mathbf{Z}}{\partial \mathbf{A}_{\mu\tau}}\frac{\partial \mathbf{A}_{\mu\tau}}{\partial \mathbf{Y}^{(\ell)}_{\alpha\beta}} = \sum_{\mu,\tau=1}^{V_{new}} (\mathbf{G}\mathbf{G}^{\top})_{\tau\mu}\frac{\partial \mathbf{A}_{\mu\tau}}{\partial \mathbf{Y}^{(\ell)}_{\alpha\beta}} \tag{S.11}$$

By considering (10), the product rule, and this key point that $\mathbf{P}^{(\ell)}$ is the only projection related to $\mathbf{Y}^{(\ell)}$, we have:

$$\frac{\partial \mathbf{A}_{\mu\tau}}{\partial \mathbf{Y}^{(\ell)}_{\alpha\beta}} = \frac{\partial \mathbf{P}^{(\ell)}_{\mu\tau}}{\partial \mathbf{Y}^{(\ell)}_{\alpha\beta}} = \delta_{\mu\beta}\sum_{i=1}^{V_{new}}\mathbf{Y}^{(\ell)}_{\tau i}\boldsymbol{\Phi}^{(\ell)}_{\alpha i} + \delta_{\tau\beta}\sum_{j=1}^{V_{new}}\mathbf{Y}^{(\ell)}_{\mu j}\boldsymbol{\Phi}^{(\ell)}_{j\alpha} + \sum_{i,j=1}^{V_{new}}\mathbf{Y}^{(\ell)}_{\mu j}\mathbf{Y}^{(\ell)}_{\tau i}\frac{\partial \boldsymbol{\Phi}^{(\ell)}_{ji}}{\partial \mathbf{Y}^{(\ell)}_{\alpha\beta}}$$

$$= \delta_{\mu\beta}\Big(\boldsymbol{\Phi}^{(\ell)}\big(\mathbf{Y}^{(\ell)}\big)^{\top}\Big)_{\alpha\tau} + \delta_{\tau\beta}\Big(\boldsymbol{\Phi}^{(\ell)}\big(\mathbf{Y}^{(\ell)}\big)^{\top}\Big)_{\alpha\mu} + \sum_{i,j=1}^{V_{new}}\mathbf{Y}^{(\ell)}_{\mu j}\mathbf{Y}^{(\ell)}_{\tau i}\frac{\partial \boldsymbol{\Phi}^{(\ell)}_{ji}}{\partial \mathbf{Y}^{(\ell)}_{\alpha\beta}} \tag{S.12}$$

The last term also can be calculated by using the chain rule as follows:

$$\frac{\partial \mathbf{\Phi}_{ji}^{(\ell)}}{\partial \mathbf{Y}_{\alpha\beta}^{(\ell)}} = \sum_{m,n=1}^{T} \frac{\partial \mathbf{\Phi}_{ji}^{(\ell)}}{\partial \widetilde{\mathbf{\Phi}}_{mn}^{(\ell)}} \frac{\widetilde{\mathbf{\Phi}}_{mn}^{(\ell)}}{\partial \mathbf{Y}_{\alpha\beta}^{(\ell)}} = -\sum_{m,n=1}^{T} \left( \mathbf{\Phi}_{jm}^{(\ell)} \mathbf{\Phi}_{ni}^{(\ell)} \left( \delta_{\alpha m} \mathbf{Y}_{\beta n}^{(\ell)} + \delta_{\alpha n} \mathbf{Y}_{\beta m}^{(\ell)} \right) \right)$$

$$= -\sum_{n=1}^{T} \mathbf{\Phi}_{j\alpha}^{(\ell)} \mathbf{\Phi}_{ni}^{(\ell)} \mathbf{Y}_{\beta n}^{(\ell)} - \sum_{m=1}^{T} \mathbf{\Phi}_{jm}^{(\ell)} \mathbf{\Phi}_{\alpha i}^{(\ell)} \mathbf{Y}_{\beta m}^{(\ell)} = -\mathbf{\Phi}_{j\alpha}^{(\ell)} \left( \mathbf{\Phi}^{(\ell)} \left( \mathbf{Y}^{(\ell)} \right)^{\top} \right)_{i\beta} - \mathbf{\Phi}_{\alpha i}^{(\ell)} \left( \mathbf{\Phi}^{(\ell)} \left( \mathbf{Y}^{(\ell)} \right)^{\top} \right)_{j\beta}$$

$$(\text{S.13})$$

By applying (S.13) to (S.12), we have:

$$\frac{\partial \mathbf{P}_{\mu\tau}^{(\ell)}}{\partial \mathbf{Y}_{\alpha\beta}^{(\ell)}} = \delta_{\mu\beta} \left( \mathbf{\Phi}^{(\ell)} \left( \mathbf{Y}^{(\ell)} \right)^{\top} \right)_{\alpha\tau} + \delta_{\tau\beta} \left( \mathbf{\Phi}^{(\ell)} \left( \mathbf{Y}^{(\ell)} \right)^{\top} \right)_{\alpha\mu} - \left( \mathbf{\Phi}^{(\ell)} \left( \mathbf{Y}^{(\ell)} \right)^{\top} \right)_{\alpha\mu} \left( \mathbf{Y}^{(\ell)} \mathbf{\Phi}^{(\ell)} \left( \mathbf{Y}^{(\ell)} \right)^{\top} \right)_{\beta\tau}$$

$$- \left( \mathbf{\Phi}^{(\ell)} \left( \mathbf{Y}^{(\ell)} \right)^{\top} \right)_{\alpha\tau} \left( \mathbf{Y}^{(\ell)} \mathbf{\Phi}^{(\ell)} \left( \mathbf{Y}^{(\ell)} \right)^{\top} \right)_{\beta\mu} = \delta_{\mu\beta} \left( \mathbf{\Phi}^{(\ell)} \left( \mathbf{Y}^{(\ell)} \right)^{\top} \right)_{\alpha\tau} + \delta_{\tau\beta} \left( \mathbf{\Phi}^{(\ell)} \left( \mathbf{Y}^{(\ell)} \right)^{\top} \right)_{\alpha\mu} -$$

$$\left( \mathbf{\Phi}^{(\ell)} \left( \mathbf{Y}^{(\ell)} \right)^{\top} \right)_{\alpha\mu} \left( \mathbf{P}^{(\ell)} \right)_{\beta\tau} - \left( \mathbf{\Phi}^{(\ell)} \left( \mathbf{Y}^{(\ell)} \right)^{\top} \right)_{\alpha\tau} \left( \mathbf{P}^{(\ell)} \right)_{\beta\mu} =$$

$$\left( \mathbf{I} - \mathbf{P}^{(\ell)} \right)_{\beta\tau} \left( \mathbf{\Phi}^{(\ell)} \left( \mathbf{Y}^{(\ell)} \right)^{\top} \right)_{\alpha\mu} + \left( \mathbf{I} - \mathbf{P}^{(\ell)} \right)_{\beta\mu} \left( \mathbf{\Phi}^{(\ell)} \left( \mathbf{Y}^{(\ell)} \right)^{\top} \right)_{\alpha\tau}$$

$$(\text{S.14})$$

Finally, we can calculate (13) as follows:

$$\frac{\partial \mathbf{Z}}{\partial \mathbf{Y}_{\alpha\beta}^{(\ell)}} = \sum_{\mu,\tau=1}^{T} \left( \left( \mathbf{G}\mathbf{G}^{\top} \right)_{\mu\tau} \left( \mathbf{I} - \mathbf{P}^{(\ell)} \right)_{\mu\beta} \left( \mathbf{\Phi}^{(\ell)} \left( \mathbf{Y}^{(\ell)} \right)^{\top} \right)_{\alpha\tau} \right)$$

$$+ \sum_{\mu,\tau=1}^{T} \left( \left( \mathbf{G}\mathbf{G}^{\top} \right)_{\mu\tau} \left( \mathbf{I} - \mathbf{P}^{(\ell)} \right)_{\tau\beta} \left( \mathbf{\Phi}^{(\ell)} \left( \mathbf{Y}^{(\ell)} \right)^{\top} \right)_{\alpha\mu} \right) = 2 \left( \mathbf{\Phi}^{(\ell)} \left( \mathbf{Y}^{(\ell)} \right)^{\top} \mathbf{G}\mathbf{G}^{\top} \left( \mathbf{I} - \mathbf{P}^{(\ell)} \right) \right)_{\alpha\beta}$$

$$(\text{S.15})$$

Consequently, we have:

$$\frac{\partial \mathbf{Z}}{\partial \mathbf{Y}^{(\ell)}} = 2\mathbf{\Phi}^{(\ell)} \left( \mathbf{Y}^{(\ell)} \right)^{\top} \mathbf{G}\mathbf{G}^{\top} \left( \mathbf{I} - \mathbf{P}^{(\ell)} \right)$$

By considering (S.10), we have:

$$\frac{\partial \mathbf{Z}}{\partial \mathbf{Y}^{(\ell)}} = 2\mathbf{R}^{(\ell)} \mathbf{G}^{\top} \left( \mathbf{I} - \mathbf{P}^{(\ell)} \right) = 2\mathbf{R}^{(\ell)} \mathbf{G}^{\top} - 2\mathbf{R}^{(\ell)} \mathbf{G}^{\top} \mathbf{P}^{(\ell)}$$

By applying (S.9), we have:

$$= 2\mathbf{R}^{(\ell)} \mathbf{G}^{\top} - 2\mathbf{R}^{(\ell)} \mathbf{G}^{\top} \mathbf{Y}^{(\ell)} \mathbf{\Phi}^{(\ell)} \left( \mathbf{Y}^{(\ell)} \right)^{\top}$$

Since $\mathbf{\Phi}^{(\ell)}$ is symmetric:

$$= 2\mathbf{R}^{(\ell)} \mathbf{G}^{\top} - 2\mathbf{R}^{(\ell)} \mathbf{G}^{\top} \mathbf{Y}^{(\ell)} \left( \mathbf{\Phi}^{(\ell)} \right)^{\top} \left( \mathbf{Y}^{(\ell)} \right)^{\top}$$

and again using (S.10), we finally have:

$$\frac{\partial \mathbf{Z}}{\partial \mathbf{Y}^{(\ell)}} = 2\mathbf{R}^{(\ell)} \mathbf{G}^{\top} - 2\mathbf{R}^{(\ell)} \left( \mathbf{R}^{(\ell)} \right)^{\top} \left( \mathbf{Y}^{(\ell)} \right)^{\top}$$

$$\implies \frac{\partial \mathbf{Z}}{\partial f_{\ell} \left( \mathbf{X}^{(\ell)}; \theta^{(\ell)} \right)} = 2\mathbf{R}^{(\ell)} \mathbf{G}^{\top} - 2\mathbf{R}^{(\ell)} \left( \mathbf{R}^{(\ell)} \right)^{\top} \left( f_{\ell} \left( \mathbf{X}^{(\ell)}; \theta^{(\ell)} \right) \right)^{\top} \blacksquare$$

## S.7 Experiment Schemes

This section presents experiment schemes. As the first step, each dataset is separately preprocessed by FSL 5.0.9 (`https://fsl.fmrib.ox.ac.uk`), i.e. slice timing, anatomical alignment, normalization, smoothing. Here, each session is preprocessed independently. Further, this paper separately applies

the anatomical alignment to each session (i.e. whole-brain image) for visualizing the neural activities in the MNI space and calibrating the Regions of Interests (ROIs). Then, ROIs are denoted by employing the main reference of each dataset. As the next step, we have partitioned preprocessed data to the training set and testing set for each iteration by using one-subject-out cross-validation. Next, the training set is applied to different HA methods. Same as the previous studies [2-7], the HA parameters are selected based on the lowest alignment error among the training data (the error is estimated by using line 10 in Algorithm 1) for both the proposed method and other methods (RHA, KHA, SRM, and CAE). Here, the structure of the deep network in the proposed method is defined based on the error of functional alignment (line 10 in Algorithm 1), i.e. activation function, regularization term, learning rate, and numbers of layers and units. As the last step in the training phase, the aligned data are employed for learning a classification (cognitive) model (without changing the alignment parameters). In the testing phase, we just applied the test data to the generated model without changing the estimated parameters of the training phase. Indeed, we try to demonstrate the effect of functional alignment on the performance of the final classification model. In other words, we consider whole of the classification procedure is fixed except the alignment section, i.e. the same preprocessed dataset for each iteration, the same learning algorithm, the same parameters for learning, etc. We have to emphasize that the same preprocessed datasets are applied to all HA methods in each iteration of the cross-validation.

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

Table S1: Variables or Functions

| Variable or Function | Description |
|---|---|
| $\mathbb{R}$ | The set of real numbers. |
| $\mathbf{I}$ | The identity matrix. |
| $i, j, \ell, m, n, \alpha, \beta, \mu, \tau$ | The indices. |
| $K$ | The number of stimulus categories. |
| $S$ | The number of Subjects. |
| $T$ | The number of time points in unites of TRs (Time of Repetition). |
| $V$ | The number of voxels in the original representation space. |
| $V_{new}$ | The number of features after applying the Artificial Neural Network (ANN). |
| $C$ | The number of layers in ANN. |
| $U^{(m)}$ | The number of units in $m$-$th$ intermediate layers of ANN. |
| $\epsilon$ | The regularized constant. |
| $\mathbf{X}^{(\ell)} = \left\{ x_{mn}^{(\ell)} \right\} \in \mathbb{R}^{T \times V}$ | The original neural activities for $\ell$-$th$ subject. |
| $\mathbf{R}^{(\ell)}$ | HA or DHA mapping for $\ell$-$th$ subject. |
| $\theta^{(\ell)} = \left\{ \mathbf{W}_m^{(\ell)}, \mathbf{b}_m^{(\ell)}, m=1:C \right\}$ | The parameters of ANN for $\ell$-$th$ subject. |
| $\mathbf{h}_m^{(\ell)} = \mathrm{g}(\dots)$ | The result of activation function for $\ell$-$th$ subject and $m$-$th$ layer. |
| $f_\ell \left( \mathbf{X}^{(\ell)}; \theta^{(\ell)} \right) = \mathbf{Y}^{(\ell)}$ | The cognitive features for $\ell$-$th$ subject after applying ANN. |
| $\mathbf{G} \in \mathbb{R}^{T \times V_{new}}$ | The DHA template. |
| $\mathbf{Q} \in \mathbb{R}^{V_{new} \times V_{new}}$ | An orthogonal matrix for mapping two different DHA template to each other. |
| $f_\ell \left( \mathbf{X}^{(\ell)}; \theta^{(\ell)} \right) \overset{SVD}{=} \mathbf{\Omega}^{(\ell)} \mathbf{\Sigma}^{(\ell)} \left( \mathbf{\Psi}^{(\ell)} \right)^\top$ | The SVD decomposition of mapped cognitive features for $\ell$-$th$ subject. |
| $\widetilde{\mathbf{\Phi}}^{(\ell)}$ | The covariance matrix for $\ell$-$th$ subject (just used in this document) |
| $\mathbf{\Phi}^{(\ell)}$ | The inverse of covariance matrix for $\ell$-$th$ subject (just used in this document) |
| $\mathbf{P}^{(\ell)}$ | The projection of cognitive features for $\ell$-$th$ subject. |
| $\mathbf{D}^{(\ell)}$ | The diagonal matrix used for decomposition of $\mathbf{P}^{(\ell)}$ for $\ell$-$th$ subject. |
| $\mathbf{A} = \widetilde{\mathbf{A}} \widetilde{\mathbf{A}}^\top$ | Sum of all projections and its Cholesky decomposition. |
| $\Lambda = \left\{ \lambda_1 \dots \lambda_T \right\}$ | Eigenvalues of $\mathbf{A}$. |
| $\mathbf{Z} = \sum_{\ell=1}^{T} \lambda_\ell$ | Sum of eigenvalues of $\mathbf{A}$. |
| $\eta$ | Learning rate. |
| $\nabla \theta^{(\ell)}$ | The gradient of ANN parameters for $\ell$-$th$ subject. |
| $\gamma_m$ | Error of $m$-$th$ iteration. |
| $k_1, k_3, \lambda, \rho$ | Convolutional Autoencoder (CAE) parameters. |
| $tr()$ | The trace function. |
| $backprop()$ | The back-propagation function. |
| $\mathrm{g}(\mathbf{x}) = \frac{1}{1+\exp(-\mathbf{x})}$ | Sigmoid function. |
| $\mathrm{g}(\mathbf{x}) = \tanh(\mathbf{x}) = \frac{\exp(\mathbf{x}) - \exp(-\mathbf{x})}{\exp(\mathbf{x}) + \exp(-\mathbf{x})}$ | Hyperbolic function. |
| $\mathrm{g}(\mathbf{x}) = \ln(1 + \exp(\mathbf{x}))$ | Rectified Linear Unit (ReLU). |

Table S2: The datasets.

| Title | ID | S | K | T | V | X | Y | Z | Scanner | TR | TE |
|---|---|---|---|---|---|---|---|---|---|---|---|
| Mixed-gambles task | DS005 | 48 | 2 | 240 | 450 | 53 | 63 | 52 | S 3T | 2 | 30 |
| Visual Object Recognition | DS105 | 71 | 8 | 121 | 1963 | 79 | 95 | 79 | G 3T | 2.5 | 30 |
| Word and Object Processing | DS107 | 98 | 4 | 164 | 932 | 53 | 63 | 52 | S 3T | 2 | 28 |
| Auditory and Visual Oddball | DS116 | 102 | 2 | 170 | 2532 | 53 | 63 | 40 | P 3T | 2 | 25 |
| Multi-subject, multi-modal | DS117 | 171 | 2 | 210 | 524 | 64 | 61 | 33 | S 3T | 2 | 30 |
| Forrest Gump | DS113 | 20 | 10 | 451 | 2400 | 160 | 160 | 36 | S 7T | 2.3 | 22 |
| Raiders of the Lost Ark | $N/A$ | 10 | 7 | 924 | 980 | 78 | 78 | 54 | S 3T | 3 | 30 |

S is the number of subject; K denotes the number of stimulus categories; T is the number of scans in unites of TRs (Time of Repetition); V denotes the number of voxels in ROI; X, Y, Z are the size of 3D images; Scanners include S=Siemens, G = General Electric, and P = Philips in 3 Tesla or 7 Tesla; TR is Time of Repetition in millisecond; TE denotes Echo Time in second; Please see *openfmri.org* for more information.

Figure S1: Visualizing average of active regions in common space across category of visual stimuli by using DS105 dataset

Figure S2: Graphical abstract of Deep Hyperalignment (DHA)



[Supplementary Material 2]

iBRAIN

# Deep Hyperalignment

**Muhammad Yousefnezhad, Daoqiang Zhang**
*31st Advances in Neural Information Processing Systems (NIPS-17)*

Dec 2017

# What is
## Hyperalignment?

# Brain Function Analysis

Smith, Nature, 2013

# Hyperalignment

Individual Brain Patterns

Common Space (G)

Subject 1

Voxel 1

Voxel 2

Subject S

Voxel 2

Voxel 1

# A Generalized Approach

$$\min_{\mathbf{R}^{(i)}, \mathbf{R}^{(j)}} \sum_{i=1}^{S} \sum_{j=i+1}^{S} \left\| f(\mathbf{X}^{(i)})\mathbf{R}^{(i)} - f(\mathbf{X}^{(j)})\mathbf{R}^{(j)} \right\|_F^2$$

$$\text{s.t.} \quad \left( \mathbf{R}^{(\ell)} \right)^\top \left( \left( f(\mathbf{X}^{(\ell)}) \right)^\top f(\mathbf{X}^{(\ell)}) + \epsilon \mathbf{I} \right) \mathbf{R}^{(\ell)} = \mathbf{I}, \quad \ell = 1{:}S$$

⭐ **If $f(\mathbf{x}) = \mathbf{x}$ & $\epsilon = 0$, then we have the original HA**

⭐ **If $f(\mathbf{x}) = \mathbf{x}$ & $\epsilon \neq 0$, then we have the Regularized HA**

⭐ **If $f(\mathbf{x})$ is a nonlinear kernel, then we have the Kernel HA**

# Deep
## Hyperalignment

# Main Idea

# DHA: Objective Function

★ **We want to optimize following function:**

$$\min_{\substack{\theta^{(i)}, \mathbf{R}^{(i)} \\ \theta^{(j)}, \mathbf{R}^{(j)}}} \sum_{i=1}^{S} \sum_{j=i+1}^{S} \left\| f_i\big(\mathbf{X}^{(i)}; \theta^{(i)}\big) \mathbf{R}^{(i)} - f_j\big(\mathbf{X}^{(j)}; \theta^{(j)}\big) \mathbf{R}^{(j)} \right\|_F^2$$

$$\mathbf{s.t.} \quad \big(\mathbf{R}^{(\ell)}\big)^\top \left( \Big( f_\ell\big(\mathbf{X}^{(\ell)}; \theta^{(\ell)}\big) \Big)^\top f_\ell\big(\mathbf{X}^{(\ell)}; \theta^{(\ell)}\big) + \epsilon \mathbf{I} \right) \mathbf{R}^{(\ell)} = \mathbf{I}, \quad \ell = 1{:}S$$

**where the deep network is defined as follows:**

$$f_\ell\big(\mathbf{X}^{(\ell)}; \theta^{(\ell)}\big) = \mathbf{mat}\Big( \mathbf{h}_C^{(\ell)}, T, V_{new} \Big)$$

$$\mathbf{h}_m^{(\ell)} = \mathbf{g}\Big( \mathbf{W}_m^{(\ell)} \mathbf{h}_{m-1}^{(\ell)} + \mathbf{b}_m^{(\ell)} \Big), \quad \mathbf{where} \quad \mathbf{h}_1^{(\ell)} = \mathbf{vec}\big(\mathbf{X}^{(\ell)}\big) \quad \mathbf{and} \quad m = 2{:}C$$

# Generalized DHA

$$\min_{\mathbf{G},\, \mathbf{R}^{(i)},\, \theta^{(i)}} \sum_{i=1}^{S} \left\| \mathbf{G} - f_i\!\left(\mathbf{X}^{(i)}; \theta^{(i)}\right) \mathbf{R}^{(i)} \right\|_F^2$$

$$\text{s.t.} \quad \mathbf{G}^\top \mathbf{G} = \mathbf{I}$$

**where**

$$\mathbf{G} = \frac{1}{S} \sum_{j=1}^{S} f_j\!\left(\mathbf{X}^{(j)}; \theta^{(j)}\right) \mathbf{R}^{(j)}$$

# DHA: Optimization

★ ***rank-m* SVD**

$$f_\ell\big(\mathbf{X}^{(\ell)};\theta^{(\ell)}\big) \overset{SVD}{=} \mathbf{\Omega}^{(\ell)}\mathbf{\Sigma}^{(\ell)}\big(\mathbf{\Psi}^{(\ell)}\big)^\top, \qquad \ell = 1{:}S$$

★ ***Projection Matrix***

$$\mathbf{P}^{(\ell)} = f_\ell\big(\mathbf{X}^{(\ell)};\theta^{(\ell)}\big)\bigg(\big(f_\ell(\mathbf{X}^{(\ell)};\theta^{(\ell)})\big)^\top f_\ell\big(\mathbf{X}^{(\ell)};\theta^{(\ell)}\big) + \epsilon\mathbf{I}\bigg)^{-1}\big(f_\ell(\mathbf{X}^{(\ell)};\theta^{(\ell)})\big)^\top$$

$$= \mathbf{\Omega}^{(\ell)}\big(\mathbf{\Sigma}^{(\ell)}\big)^\top\big(\mathbf{\Sigma}^{(\ell)}\big(\mathbf{\Sigma}^{(\ell)}\big)^\top + \epsilon\mathbf{I}\big)^{-1}\mathbf{\Sigma}^{(\ell)}\big(\mathbf{\Omega}^{(\ell)}\big)^\top = \mathbf{\Omega}^{(\ell)}\mathbf{D}^{(\ell)}\big(\mathbf{\Omega}^{(\ell)}\mathbf{D}^{(\ell)}\big)^\top$$

***where*** $\mathbf{D}^{(\ell)}\big(\mathbf{D}^{(\ell)}\big)^\top = \big(\mathbf{\Sigma}^{(\ell)}\big)^\top\big(\mathbf{\Sigma}^{(\ell)}\big(\mathbf{\Sigma}^{(\ell)}\big)^\top + \epsilon\mathbf{I}\big)^{-1}\mathbf{\Sigma}^{(\ell)}\,.$

★ **Sum of *Projection Matrices***

$$\mathbf{A} = \sum_{i=1}^{S}\mathbf{P}^{(i)} = \widetilde{\mathbf{A}}\,\widetilde{\mathbf{A}}^\top, \quad \textbf{where} \quad \widetilde{\mathbf{A}} \in \mathbb{R}^{T \times mS} = \big[\mathbf{\Omega}^{(1)}\mathbf{D}^{(1)}\ldots\mathbf{\Omega}^{(S)}\mathbf{D}^{(S)}\big]\,.$$

**Cholesky Decomposition**

# DHA: Optimization

★ *Objective Function can be reformulated as follows:*

$$\min_{\mathbf{G}, \mathbf{R}^{(i)}, \theta^{(i)}} \sum_{i=1}^{S} \left\| \mathbf{G} - f_i\big(\mathbf{X}^{(i)}; \theta^{(i)}\big) \mathbf{R}^{(i)} \right\| \equiv \max_{\mathbf{G}} \left( \mathbf{tr}\big(\mathbf{G}^{\top} \mathbf{A} \mathbf{G}\big) \right).$$

★ *So, we have:*

$$\mathbf{AG} = \mathbf{G}\Lambda, \text{ where } \Lambda = \left\{ \lambda_1 \dots \lambda_T \right\}$$

$$\widetilde{\mathbf{A}} = \mathbf{G} \widetilde{\Sigma} \widetilde{\Psi}^{\top} \longrightarrow \text{ Incremental PCA}$$

★ *DHA mappings can be calculated as follows:*

$$\mathbf{R}^{(\ell)} = \left( \left( f_\ell\big(\mathbf{X}^{(\ell)}; \theta^{(\ell)}\big) \right)^{\top} f_\ell\big(\mathbf{X}^{(\ell)}; \theta^{(\ell)}\big) + \epsilon \mathbf{I} \right)^{-1} \left( f_\ell\big(\mathbf{X}^{(\ell)}; \theta^{(\ell)}\big) \right)^{\top} \mathbf{G}.$$

# DHA: Optimization

⭐ *In order to use back-propagation algorithm for seeking an optimized parameters for the deep network, we also have:*

$$\frac{\partial \mathbf{Z}}{\partial f_\ell\left(\mathbf{X}^{(\ell)};\theta^{(\ell)}\right)} = 2\mathbf{R}^{(\ell)}\mathbf{G}^\top - 2\mathbf{R}^{(\ell)}\left(\mathbf{R}^{(\ell)}\right)^\top \left(f_\ell\left(\mathbf{X}^{(\ell)};\theta^{(\ell)}\right)\right)^\top.$$

**where**

$$\mathbf{Z} = \sum_{\ell=1}^{T} \lambda_\ell$$

# Empirical Studies

# Datasets

Table S2: The datasets.

| Title | ID | S | K | T | V | X | Y | Z | Scanner | TR | TE |
|---|---|---|---|---|---|---|---|---|---|---|---|
| Mixed-gambles task | DS005 | 48 | 2 | 240 | 450 | 53 | 63 | 52 | S 3T | 2 | 30 |
| Visual Object Recognition | DS105 | 71 | 8 | 121 | 1963 | 79 | 95 | 79 | G 3T | 2.5 | 30 |
| Word and Object Processing | DS107 | 98 | 4 | 164 | 932 | 53 | 63 | 52 | S 3T | 2 | 28 |
| Auditory and Visual Oddball | DS116 | 102 | 2 | 170 | 2532 | 53 | 63 | 40 | P 3T | 2 | 25 |
| Multi-subject, multi-modal | DS117 | 171 | 2 | 210 | 524 | 64 | 61 | 33 | S 3T | 2 | 30 |
| Forrest Gump | DS113 | 20 | 10 | 451 | 2400 | 160 | 160 | 36 | S 7T | 2.3 | 22 |
| Raiders of the Lost Ark | $N/A$ | 10 | 7 | 924 | 980 | 78 | 78 | 54 | S 3T | 3 | 30 |

S is the number of subject; K denotes the number of stimulus categories; T is the number of scans in unites of TRs (Time of Repetition); V denotes the number of voxels in ROI; X, Y, Z are the size of 3D images; Scanners include S=Siemens, G = General Electric, and P = Philips in 3 Tesla or 7 Tesla; TR is Time of Repetition in millisecond; TE denotes Echo Time in second; Please see *openfmri.org* for more information.

# Simple Tasks Analysis

Table 1: Accuracy of HA methods in post-alignment classification by using simple task datasets

| ↓Algorithms, Datasets→ | DS005 | DS105 | DS107 | DS116 | DS117 |
|---|---|---|---|---|---|
| $\nu$-SVM [17] | 71.65±0.97 | 22.89±1.02 | 38.84±0.82 | 67.26±1.99 | 73.32±1.67 |
| HA [1] | 81.27±0.59 | 30.03±0.87 | 43.01±0.56 | 74.23±1.40 | 77.93±0.29 |
| RHA [2] | 83.06±0.36 | 32.62±0.52 | 46.82±0.37 | 78.71±0.76 | 84.22±0.44 |
| KHA [3] | 85.29±0.49 | 37.14±0.91 | 52.69±0.69 | 78.03±0.89 | 83.32±0.41 |
| SVD-HA [4] | 90.82±1.23 | 40.21±0.83 | 59.54±0.99 | 81.56±0.54 | 95.62±0.83 |
| SRM [5] | 91.26±0.34 | 48.77±0.94 | 64.11±0.37 | 83.31±0.73 | 95.01±0.64 |
| SL [9] | 90.21±0.61 | 49.86±0.4 | 64.07±0.98 | 82.32±0.28 | 94.96±0.24 |
| CAE [6] | 94.25±0.76 | 54.52±0.80 | 72.16±0.43 | **91.49±0.67** | 95.92±0.67 |
| DHA | **97.92±0.82** | **60.39±0.68** | **73.05±0.63** | 90.28±0.71 | **97.99±0.94** |

Table 2: Area under the ROC curve (AUC) of different HA methods in post-alignment classification by using simple task datasets

| ↓Algorithms, Datasets→ | DS005 | DS105 | DS107 | DS116 | DS117 |
|---|---|---|---|---|---|
| $\nu$-SVM [17] | 68.37±1.01 | 21.76±0.91 | 36.84±1.45 | 62.49±1.34 | 70.17±0.59 |
| HA [1] | 70.32±0.92 | 28.91±1.03 | 40.21±0.33 | 70.67±0.97 | 76.14±0.49 |
| RHA [2] | 82.22±0.42 | 30.35±0.39 | 43.63±0.61 | 76.34±0.45 | 81.54±0.92 |
| KHA [3] | 80.91±0.21 | 36.23±0.57 | 50.41±0.92 | 75.28±0.94 | 80.92±0.28 |
| SVD-HA [4] | 88.54±0.71 | 37.61±0.62 | 57.54±0.31 | 78.66±0.82 | 92.14±0.42 |
| SRM [5] | 90.23±0.74 | 44.48±0.75 | 62.41±0.72 | 79.20±0.98 | 93.65±0.93 |
| SL [9] | 89.79±0.25 | 47.32±0.92 | 61.84±0.32 | 80.63±0.81 | 93.26±0.72 |
| CAE [6] | 91.24±0.61 | 52.16±0.63 | **72.33±0.79** | 87.53±0.72 | 91.49±0.33 |
| DHA | **96.91±0.82** | **59.57±0.32** | 70.23±0.92 | **89.93±0.24** | **96.13±0.32** |

(a) Forrest Gump (TRs = 100)

(b) Forrest Gump (TRs = 400)

(c) Forrest Gump (TRs = 800)

(d) Forrest Gump (TRs = 2000)

(e) Raiders (TRs = 100)

(f) Raiders (TRs = 400)

(g) Raiders (TRs = 800)

(h) Raiders (TRs = 2000)

Figure 1: Comparison of different HA algorithms on complex task datasets by using ranked voxels.

# Classification analysis by using feature selection

(A) DS105&emsp;&emsp;&emsp;&emsp;&emsp;&emsp;(B) DS107

(A) DS105

(B) DS107

# Future
## Works

# Future Works

* ★ **This paper extended a deep approach for hyperalignment methods in order to provide accurate functional alignment in multi-subject fMRI analysis.**

* ★ **Deep Hyperalignment (DHA) can handle fMRI datasets with nonlinearity, high-dimensionality (broad ROI), and a large number of subjects. Further, its time complexity fairly scales with data size and the training data is not referenced when DHA computes the functional alignment for a new subject.**

* ★ **In the future, we will plan to employ DHA for improving the performance of other techniques in fMRI analysis, e.g. Representational Similarity Analysis (RSA).**

Deep Hyperalignment

# Thank You!

# Q & A

For more details, contact:

myousefnezhad@nuaa.edu.cn

myousefnezhad@outlook.com

https://myousefnezhad.github.io/


[Supplementary Material 3]

# Deep Hyperalignment

**iBRAIN**

*Muhammad Yousefnezhad, Daoqiang Zhang*

School of Computer Science and Technology, Nanjing University of Aeronautics and Astronautics, Nanjing, China

## MOTIVATION

★ **A generalized approach for classical Hyperalignment (HA):**

$$\min_{\mathbf{R}^{(i)},\mathbf{R}^{(j)}} \sum_{i=1}^{S} \sum_{j=i+1}^{S} \left\| f(\mathbf{X}^{(i)})\mathbf{R}^{(i)} - f(\mathbf{X}^{(j)})\mathbf{R}^{(j)} \right\|_F^2$$

$$\text{s.t.} \quad \left(\mathbf{R}^{(\ell)}\right)^\top \left( \left(f(\mathbf{X}^{(\ell)})\right)^\top f(\mathbf{X}^{(\ell)}) + \epsilon\mathbf{I} \right)\mathbf{R}^{(\ell)} = \mathbf{I}, \quad \ell = 1:S$$

**If** $f(\mathbf{x}) = \mathbf{x}$ & $\epsilon = 0$, then we have the original HA

**If** $f(\mathbf{x}) = \mathbf{x}$ & $\epsilon \neq 0$, then we have the Regularized HA

**If** $f(\mathbf{x})$ is a nonlinear kernel, then we have the Kernel HA

| | |
|---|---|
| $\mathbf{X}^{(\ell)}$: | denotes brain activities |
| $\mathbf{R}^{(\ell)}$: | is DHA mapping |

➡ **These methods are limited by a restricted fixed kernel function.**

## METHOD

**Graphical abstract of Deep Hyperalignment (DHA)**

★ **Objective Function:**

$$\min_{\mathbf{G},\mathbf{R}^{(i)},\theta^{(i)}} \sum_{i=1}^{S} \left\| \mathbf{G} - f_i(\mathbf{X}^{(i)};\theta^{(i)})\mathbf{R}^{(i)} \right\|_F^2 \qquad \text{s.t.} \quad \mathbf{G}^\top\mathbf{G} = \mathbf{I}$$

**Shared Space:** $\mathbf{G} = \dfrac{1}{S}\sum_{j=1}^{S} f_j(\mathbf{X}^{(j)};\theta^{(j)})\mathbf{R}^{(j)}$

| | |
|---|---|
| $\theta^{(\ell)}$: | Network Parameters |
| $g()$: | Nonlinear Function |
| $V_{new}$: | # of voxels after mapping |
| $C$: | # of hidden layers |
| $T$: | Timepoints |

**where Deep network is defined as follows:**

$$f_\ell(\mathbf{X}^{(\ell)};\theta^{(\ell)}) = \mathbf{mat}\left(\mathbf{h}_C^{(\ell)}, T, V_{new}\right)$$

$$\mathbf{h}_m^{(\ell)} = \mathbf{g}\left(\mathbf{W}_m^{(\ell)}\mathbf{h}_{m-1}^{(\ell)} + \mathbf{b}_m^{(\ell)}\right), \quad \text{where} \quad \mathbf{h}_1^{(\ell)} = \mathbf{vec}(\mathbf{X}^{(\ell)}) \quad \text{and} \quad m = 2:C$$

★ **Optimization:**

**STEP 1: DHA shared space can be calculated as follows:**

$$\min_{\mathbf{G},\mathbf{R}^{(i)},\theta^{(i)}} \sum_{i=1}^{S} \left\| \mathbf{G} - f_i(\mathbf{X}^{(i)};\theta^{(i)})\mathbf{R}^{(i)} \right\| \equiv \max_{\mathbf{G}}\left(\mathbf{tr}\left(\mathbf{G}^\top\mathbf{A}\mathbf{G}\right)\right)$$

**where A is sum of projection matrices:**

$$\mathbf{P}^{(\ell)} = f_\ell(\mathbf{X}^{(\ell)};\theta^{(\ell)})\left(\left(f_\ell(\mathbf{X}^{(\ell)};\theta^{(\ell)})\right)^\top f_\ell(\mathbf{X}^{(\ell)};\theta^{(\ell)}) + \epsilon\mathbf{I}\right)^{-1}\left(f_\ell(\mathbf{X}^{(\ell)};\theta^{(\ell)})\right)^\top \qquad \mathbf{A} = \sum_{\ell}\mathbf{P}^{(\ell)}$$

**STEP 2: DHA mappings can be calculated as follows:**

$$\mathbf{R}^{(\ell)} = \left(\left(f_\ell(\mathbf{X}^{(\ell)};\theta^{(\ell)})\right)^\top f_\ell(\mathbf{X}^{(\ell)};\theta^{(\ell)}) + \epsilon\mathbf{I}\right)^{-1}\left(f_\ell(\mathbf{X}^{(\ell)};\theta^{(\ell)})\right)^\top \mathbf{G}.$$

**STEP 3: Deep Network parameters can be iteratively calculated as follows by using SGD:**

$$\theta^{(\ell)} \leftarrow \theta^{(\ell)} - \eta\nabla\theta^{(\ell)}, \quad \text{where} \quad \mathbf{Z} = \sum_{\ell=1}^{T}\lambda_\ell \quad \text{and}$$

$$\frac{\partial\mathbf{Z}}{\partial f_\ell(\mathbf{X}^{(\ell)};\theta^{(\ell)})} = 2\mathbf{R}^{(\ell)}\mathbf{G}^\top - 2\mathbf{R}^{(\ell)}\left(\mathbf{R}^{(\ell)}\right)^\top\left(f_\ell(\mathbf{X}^{(\ell)};\theta^{(\ell)})\right)^\top.$$

## DATASETS

★ **This paper utilizes 7 datasets for running empirical studies:**

| Title | ID | S | K | T | V | X | Y | Z | Scanner | TR | TE |
|---|---|---|---|---|---|---|---|---|---|---|---|
| Mixed-gambles task | DS005 | 48 | 2 | 240 | 450 | 53 | 63 | 52 | S 3T | 2 | 30 |
| Visual Object Recognition | DS105 | 71 | 8 | 121 | 1963 | 79 | 95 | 79 | G 3T | 2.5 | 30 |
| Word and Object Processing | DS107 | 98 | 4 | 164 | 932 | 53 | 63 | 52 | S 3T | 2 | 28 |
| Auditory and Visual Oddball | DS116 | 102 | 2 | 170 | 2532 | 53 | 63 | 40 | P 3T | 2 | 25 |
| Multi-subject, multi-modal | DS117 | 171 | 2 | 210 | 524 | 64 | 61 | 33 | S 3T | 2 | 30 |
| Forrest Gump | DS113 | 20 | 10 | 451 | 2400 | 160 | 160 | 36 | S 7T | 2.3 | 22 |
| Raiders of the Lost Ark | N/A | 10 | 7 | 924 | 980 | 78 | 78 | 54 | S 3T | 3 | 30 |

S is the number of subject; K denotes the number of stimulus categories; T is the number of scans in unites of TRs (Time of Repetition); V denotes the number of voxels in ROI; X, Y, Z are the size of 3D images; Scanners include S=Siemens, G = General Electric, and P = Philips in 3 Tesla or 7 Tesla; TR is Time of Repetition in millisecond; TE denotes Echo Time in second; Please see *openfmri.org* for more information.

## EXPERIMENTAL RESULTS

★ **Simple Tasks Analysis:**

Table 1: Accuracy of HA methods in post-alignment classification by using simple task datasets

| ↓Algorithms, Datasets→ | DS005 | DS105 | DS107 | DS116 | DS117 |
|---|---|---|---|---|---|
| $\nu$-SVM [17] | 71.65±0.97 | 22.89±1.02 | 38.84±0.82 | 67.26±1.99 | 73.32±1.67 |
| HA [1] | 81.27±0.59 | 30.03±0.87 | 43.01±0.56 | 74.23±1.40 | 77.93±0.29 |
| RHA [2] | 83.06±0.36 | 32.62±0.52 | 46.82±0.37 | 78.71±0.76 | 84.22±0.44 |
| KHA [3] | 85.29±0.49 | 37.14±0.91 | 52.69±0.69 | 78.03±0.89 | 83.32±0.41 |
| SVD-HA [4] | 90.82±1.23 | 40.21±0.83 | 59.54±0.99 | 81.56±0.54 | 95.62±0.83 |
| SRM [5] | 91.26±0.34 | 48.77±0.94 | 64.11±0.37 | 83.31±0.73 | 95.01±0.64 |
| SL [9] | 90.21±0.61 | 49.86±0.4 | 64.07±0.98 | 82.32±0.28 | 94.96±0.24 |
| CAE [6] | 94.25±0.76 | 54.52±0.80 | 72.16±0.43 | **91.49±0.67** | 95.92±0.67 |
| DHA | **97.92±0.82** | **60.39±0.68** | **73.05±0.63** | 90.28±0.71 | **97.99±0.94** |

Table 2: Area under the ROC curve (AUC) of different HA methods in post-alignment classification by using simple task datasets

| ↓Algorithms, Datasets→ | DS005 | DS105 | DS107 | DS116 | DS117 |
|---|---|---|---|---|---|
| $\nu$-SVM [17] | 68.37±1.01 | 21.76±0.91 | 36.84±1.45 | 62.49±1.34 | 70.17±0.59 |
| HA [1] | 70.32±0.92 | 28.91±1.03 | 40.21±0.33 | 70.67±0.97 | 76.14±0.49 |
| RHA [2] | 82.22±0.42 | 30.35±0.39 | 43.63±0.61 | 76.34±0.45 | 81.54±0.92 |
| KHA [3] | 80.91±0.21 | 36.23±0.57 | 50.41±0.92 | 75.28±0.94 | 80.92±0.28 |
| SVD-HA [4] | 88.54±0.71 | 37.61±0.62 | 57.54±0.31 | 78.66±0.82 | 92.14±0.42 |
| SRM [5] | 90.23±0.74 | 44.48±0.75 | 62.41±0.72 | 79.20±0.98 | 93.65±0.93 |
| SL [9] | 89.79±0.25 | 47.32±0.92 | 61.84±0.32 | 80.63±0.81 | 93.26±0.72 |
| CAE [6] | 91.24±0.61 | 52.16±0.63 | **72.33±0.79** | 87.53±0.72 | 91.49±0.33 |
| DHA | **96.91±0.82** | **59.57±0.32** | 70.23±0.92 | **89.93±0.24** | **96.13±0.32** |

★ **Complex Tasks Analysis:**

(a) Forrest Gump (TRs = 100)   (b) Forrest Gump (TRs = 400)   (c) Forrest Gump (TRs = 800)   (d) Forrest Gump (TRs = 2000)

(e) Raiders (TRs = 100)   (f) Raiders (TRs = 400)   (g) Raiders (TRs = 800)   (h) Raiders (TRs = 2000)

★ **Runtime Analysis:**

(A) DS105    (B) DS107

## CONCLUSION

This paper extended a deep approach for hyperalignment methods in order to provide accurate functional alignment in multi-subject fMRI analysis. Deep Hyperalignment (DHA) can handle fMRI datasets with nonlinearity, high-dimensionality (broad ROI), and a large number of subjects. DHA is parametric and uses rank-$m$ SVD and stochastic gradient descent for optimization. Experimental studies on multi-subject fMRI datasets confirm that the DHA method achieves superior performance to other state-of-the-art HA algorithms. In the future, we will plan to employ DHA for improving the performance of other techniques in fMRI analysis, e.g. Representational Similarity Analysis (RSA), multi-modality, and hub detection.