[Reviews · NeurIPS 2017]

Reviewer 1



Summary This paper proposes a hyperalignment method based on using neural network in place of basis functions under a framework similar to kernel hyperalignment. The method is applied to a number of real fMRI datasets. Increased classification accuracy with reduced computation time compared to another deep learning architecture is shown. Comment 1. The novelty of this paper is unclear. From the text, the proposed method seems to be a combination of previous contributions. Does using neural network in place of basis functions result in an objective that is much harder to optimize? The optimization procedure seems to be “inspired” by previous works also. 2. Works on functional alignment by Georg Lang should be cited. 3. The original purpose of alignment is to establish subject correspondence so subject variability can be modelled. The best way to demonstrate better correspondence is to visually show that the subjects’ activation patterns better align. By using a classification task for assessment as done in this work, it is difficulty to tell if it is better alignment that resulted in better classification or simply a better functional representation that resulted in better performance. In fact, it is unclear how to map the aligned data to the original anatomical space for interpretation purposes. 4. Since G is not unique, would that have a major impact on the results? 5. The results might reflect overfitting, i.e. biased, since the best performance over a range of parameter values is reported, as opposed to conducting proper nested cross validation. Models with greater flexibility would tend to perform better. Also, the range of parameter values would have a great influence on the results. 6. Functional hyperalignment presumably does not require pre-anatomical alignment. Does the performance gain hold if no spatial normalization is performed?

Reviewer 2



This work describes an extension of the hyperalignment (HA) method, i.e. functional alignment across subjects. This extension directly addresses some limitations of previous HA methods in the literature: nonlinearity, scalability and high-dimensionality. The proposed method, called deep hyperalignment (DHA), defines the objective function to minimize by reformulating the HA objective function in terms of multiple stacked layers of nonlinear transformations with a (multi-layer) kernel function. The objective function is reformulated in terms of templates and then eigendecomposition. Explicit derivative is derived. Thus, optimization can be carried out also for a dataset with large number of subjects. The proposed method is tested using standard SVM classification and compared to the main HA algorithms in the literature. The experimental setup is based on 5 task-based datasets and 2 movie-based datasets. DHA shows superior performance with respect to other algorithms, in all cases. The paper is well written, even though complex to read in Section 3 ("Deep Hyperalignment"). I invite the authors to make that part more accessible. I see no major issues in the paper, but I am not expert in the mathematical derivations of the proposed method. The experimental evidence is convincing and significant.

Reviewer 3



Deep Hyperalignment [Due to time constraints, I did not read the entire paper in detail.] The problem of functional alignment of multi-subject fMRI datasets is a fundamental and difficult problem in neuroimaging. Hyperalignment is a prominent class of anatomy-free functional alignment methods. The present manuscript presents a extension to this using a deep NN kernel. I believe this to be novel, though I don’t know the literature well enough to be confident of that. While not a surprising research direction, this is a natural and promising method to consider, and the experimental results are excellent. The methods are explained thoroughly, with detailed proofs in supplementary material. I did however find this section too dense and without a good summary of the final design and parameter choices. A figure sketching the method would have been very helpful. Related work is surveyed well. The new method could be described as an amalgam of several existing approaches. However the various elements have been combined thoughtfully in a novel way, and carefully implemented. The experiments are impressively thorough, involving several open datasets and several competing methods, with detailed reporting of methods, including the parameters used for each of the alignment methods. Some more detail on preprocessing would have been helpful. It seems there has been some validation of parameters on the test data (“the best performance for each dataset is reported” p6), which is not ideal, however the number of parameter values tested is small and the results are so good that I feel they would stand up to a better training/validation/test split. My main concern about this section is that all of the results reported were computed by the present authors; it would be helpful to also flag any comparable results in the literature. The quality of the English is mostly good but with occasional problems. E.g “time complexity fairly scales with data size” is unclear, and the sentence in which it appears is repeated three times in the manuscript (including once in the abstract). Some proof-reading is in order.